# Inhibition of DUSP18 impairs cholesterol biosynthesis and promotes anti-tumor immunity in colorectal cancer

Xiaojun Zhou [1,2], Genxin Wang[1,2], Chenhui Tian[1,2], Lin Du[1,2], Edward V. Prochownik[3,4,5] & Youjun Li [1,2] ✉

Tumor cells reprogram their metabolism to produce specialized metabolites that both fuel their own growth and license tumor immune evasion. However, the relationships between these functions remain poorly understood. Here, we report CRISPR screens in a mouse model of colo-rectal cancer (CRC) that implicates the dual specificity phosphatase 18 (DUSP18) in the establishment of tumor-directed immune evasion. *Dusp18* inhibition reduces CRC growth rates, which correlate with high levels of CD8+ T cell activation. Mechanistically, DUSP18 dephosphorylates and stabilizes the USF1 bHLH-ZIP transcription factor. In turn, USF1 induces the *SREBF2* gene, which allows cells to accumulate the cholesterol biosynthesis intermediate lanosterol and release it into the tumor microenvironment (TME). There, lanosterol uptake by CD8+ T cells suppresses the mevalonate pathway and reduces KRAS protein prenylation and function, which in turn inhibits their activation and establishes a molecular basis for tumor cell immune escape. Finally, the combination of an anti-PD-1 antibody and Lumacaftor, an FDA-approved small molecule inhibitor of DUSP18, inhibits CRC growth in mice and synergistically enhances anti-tumor immunity. Collectively, our findings support the idea that a combination of immune checkpoint and metabolic blockade represents a rationally-designed, mechanistically-based and potential therapy for CRC.

World-wide, colorectal cancer (CRC) is the third most prevalent and the second most lethal malignancy[1–3], the current treatment of which consists of surgical resection and chemotherapy[4]. With the initial success of melanoma and lung cancer treatment, immunotherapy has rapidly become a major treatment option for many solid cancers, including certain molecular subtypes of CRC[5,6]. However, only about 15% of CRC patients currently benefit from immune checkpoint blockade (ICB) therapy[6]. One reason for this low response rate is that tumors remodel their microenvironment in ways that promote the exhaustion and inactivation of infiltrating CD8+ T cells, thereby leading to "immune escape". CD8+ T cells initially infiltrate tumors and speci-fically recognize tumor antigens in order to initiate killing[5]. However, tumor cells can counter this by contributing to the formation of a variety of immunosuppressive tumor microenvironments (TMEs)[7,8]. These can limit the infiltration, activation and cytotoxicity of CD8+ T cells by reducing the display of MHC-I molecules on tumor cells[9], suppressing IFN signaling[10], repressing chemokine production[11], altering the composition of the extracellular matrix[12], and increasing

[1]Department of Colorectal and Anal Surgery, Zhongnan Hospital of Wuhan University, Hubei Key Laboratory of Cell Homeostasis, College of Life Sciences, Wuhan University, Wuhan, Hubei 430072, China. [2]Frontier Science Center for Immunology and Metabolism, Medical Research Institute, TaiKang Center for Life and Medical Sciences, Wuhan University, Wuhan, Hubei 430071, China. [3]Division of Hematology/Oncology, Children's Hospital of Pittsburgh of UPMC, Pittsburgh, PA 15224, USA. [4]Department of Microbiology and Molecular Genetics of UPMC, Pittsburgh, PA 15224, USA. [5]The Pittsburgh Liver Research Center, The Hillman Cancer Institute of UPMC, Pittsburgh, PA 15224, USA. ✉e-mail: liy7@whu.edu.cn

the expression of co-inhibitory molecules such as PD-L1[13,14]. For example, TRIB3 can reduce CD8+ T cell infiltration and induce immune evasion by inhibiting the STAT1-CXCL10 axis in CRC[15]. Loss of mitochondrial electron transport chain complex II has also been shown to increase antigen presentation and T cell-mediated killing[16]. Oncogenic KRAS in tumor cells can inhibit the expression of IRF2, leading to high expression of CXCL3, which promotes the migration of myeloid-derived stem cells into the TME[17]. Finally, the down-regulation of tumor ACSL4 can inhibit ferroptosis that is induced by cytotoxic T lymphocytes (CTLs)[18].

Dual-specificity phosphatases (DUSPs) are heterotrimeric protein phosphatases that dephosphorylate tyrosine (Tyr), serine (Ser) and threonine (Thr) residues and, regulate many important pathways pertinent to metabolism, carcinogenesis, epigenetic modeling and survival. For example, *PTPN2/PTPN1* inhibition promotes the function of natural killer (NK) cells and CD8+ T cells in the TME by augmenting JAK-STAT signaling and reducing T cell dysfunction[19]. The phosphatase PAC1 acts as a T-cell suppressor that weakens host antitumor immunity[20]. *MKP5*-deficient T cells express higher levels of pro-inflammatory cytokines during innate immune responses and exhibited greater T-cell activation[21].

DUSP18, a little-studied phosphatase, has previously been reported to catalyze MAPK14 dephosphorylation, thereby inhibiting TP53 phosphorylation and functionally contributing to the malignant behavior of hepatocellular carcinoma cells[22]. However, it is not known whether DUSP18 regulates CRC antitumor immunity.

Tumors can rewire their metabolism to suppress antitumor immunity[23]. For example, excessive fumarate, ammonia and linoleic acid generated by tumor cells can accumulate in the TME, suppress the infiltration and activation of CD8+ T cells and thus minimize their antitumor effects[24–26]. Elucidating the mechanisms by which tumors and their various products can suppress T cell infiltration and activation are therefore essential for improving both innate and therapy-directed immune responses[27].

Cholesterol is essential for cell proliferation and migration while also serving as a signaling molecule in cancer. Its biosynthesis requires the concerted and highly controlled action of more than 20 enzymes[28–31]. Various cholesterol intermediates, end-products and cholesterol-derived metabolites play important roles in cellular metabolic homeostasis and remodeling of the TME[32]. For instance, PCSK9 regulates the expression of MHC-I on the tumor cells and its inhibition promotes robust cytotoxic T cell infiltration[33,34]. Inhibition of the cholesterol-esterification enzyme ACAT1 reprograms cholesterol metabolism in CD8+ T cells and leads to the accumulation of free cholesterol at the plasma membrane[35]. Cholesterol release by tumor cells into the TME can drive immune-suppressive reprogramming by activating bone marrow-derived suppressor cells[36]. 27-hydroxycholesterol can alter the LXR and SREBP2 pathways so as to deplete cholesterol reserves and drive T-cell exhaustion and dysfunction[37]. 7-dehydrocholesterol can regulate type I interferon (IFN) production by modulating AKT3 activation[38].

In this work, we employ CRISPR screens and identify DUSP18 as a factor that limits the activation of CD8+ T cells and their ability to suppress CRC growth. Mechanically, DUSP18 increases the activity of the USF1-SREBP2 transcription factor (TF) axis, upregulates the cholesterol biosynthetic pathway and allows for the accumulation of lanosterol, a cholesterol precursor, in cancer cells. CRC cells release lanosterol into the TME where its uptake by CD8+ T cells inhibits the mevalonate pathway, leading to reduced KRAS prenylation, inhibition of downstream KRAS signaling and ultimately CD8+ T cell inactivation. Together, our findings reveal a metabolic role of DUSP18 in the promotion of immune evasion. We further show that the combination of DUSP18 inhibition and immune checkpoint blockade enhances the antitumor activity of CD8+ T cells in a mouse model of CRC. This suggests a potential form of combination therapy that is rationally designed and based on the targeting of both metabolic and immune factors so as to activate and sustain the antitumor activity of CD8+ T cells.

## Results

### Inhibition of DUSP18 enhances the function of tumor-infiltrating CD8+ T cells

To systematically identify genetic targets in cancer cells whose loss represses immune evasion, a method to identify these genes was developed (Fig. 1a). MC38 CRC cells were engineered to express Cas9 (Supplementary Fig. 1a) and single guide RNAs (sgRNAs) from a Mouse CRISPR Deletion Library (termed KPD library) generated against ~2000 genes encoding drug targets, kinases and phosphatases (Supplementary Data 1)[39]. After in vitro puromycin selection for 7 days to ensure stable gene editing, the tumor cells were subcutaneously transplanted into immunodeficient or immunocompetent mice, with the latter being intended to generate an immune-selective pressure on the tumor cells. At the time of harvesting 16 days later, tumors grown in immunodeficient hosts were significantly larger (Supplementary Fig. 1b). Upon amplification and sequencing of integrated sgRNA inserts from each group, we identified a number that were depleted in tumors from the immunocompetent cohort based on the MAGeCK algorithm (Fig. 1b and Supplementary Data 2). As evidence that certain gRNAs could be predictably selected against, those targeting *Ptgs1* (*Cox1*), which promotes PGE2 production and decreases CD8+ T cell infiltration, were depleted in tumors engrafted in immunocompetent mice[40]. A number of other methods including DrugZ[41] and DESeq2[42] for analyzing CRISPR screens results were also adopted to verify the accuracy of our analysis (Supplementary Fig. 1c and Supplementary Data 2). The results showed that some genes with the potential to promote tumor immune evasion, such as *Ptgs1* and *Dusp18*, etc., were significantly enriched in all three analytical methods, which excludes the effect of different analytical methods on the differences in results.

The Cancer Genome Atlas (TCGA)-colon adenocarcinoma (COAD) dataset was also analyzed to identify cytotoxic T lymphocyte (CTL) scores using a set of 5 previously reported genes (*CD8A*, *CD8B*, *GZMA*, *GZMB*, and *PRF1*) as a surrogate of tumor infiltrating CD8+ T cells[43]. This analysis revealed that patients with higher CTL scores experienced better survival (Supplementary Fig. 1d). Next, the correlation between protein-coding genes and CTL scores was calculated and the results showed that 735 genes were negatively associated with CTL scores (Supplementary Data 3). Finally, the intersection between the CRISPR screens selected genes and those correlating negatively with CTL scores was determined with *Dusp18* being the most noteworthy (Fig. 1c). Higher *DUSP18* mRNA expression level was also found in CRC samples compared with normal colo-rectal tissues (Fig. 1d). *DUSP18* expression also negatively correlated with 3 of the genes used to calculate the CTL scores, namely *CD8A*, *PRF1* and *GZMA* as well as 3 others (*TNFRSF18*, *GZMH* and *GNLY*) (Fig. 1e), while also correlating with CD8+ T cell infiltration (Supplementary Fig. 1e). Kaplan–Meier survival curves based on high or low *DUSP18* mRNA expression were generated using the Tumor Immune Dysfunction and Exclusion (TIDE) tool[43] to explore the association between CTL scores and overall survival in CRC patients. In those CRC patients with low *DUSP18* mRNA expression, high CTL scores were associated with better survival whereas those with high *DUSP18* mRNA expression and high CTL scores showed worse survival. Taken together, these findings suggest that high *DUSP18* mRNA expression is associated with T cell dysfunction (Supplementary Fig. 1f).

To determine whether *Dusp18* inhibition affects T cell-mediated antitumor function, shRNA-mediated inhibition of *Dusp18* was performed in MC38 CRC cells and ovalbumin (OVA)-expressing B16 melanoma (B16-OVA) cells (Supplementary Fig. 1g, h). This did not impair tumor cell proliferation in vitro (Supplementary Fig. 1i) or

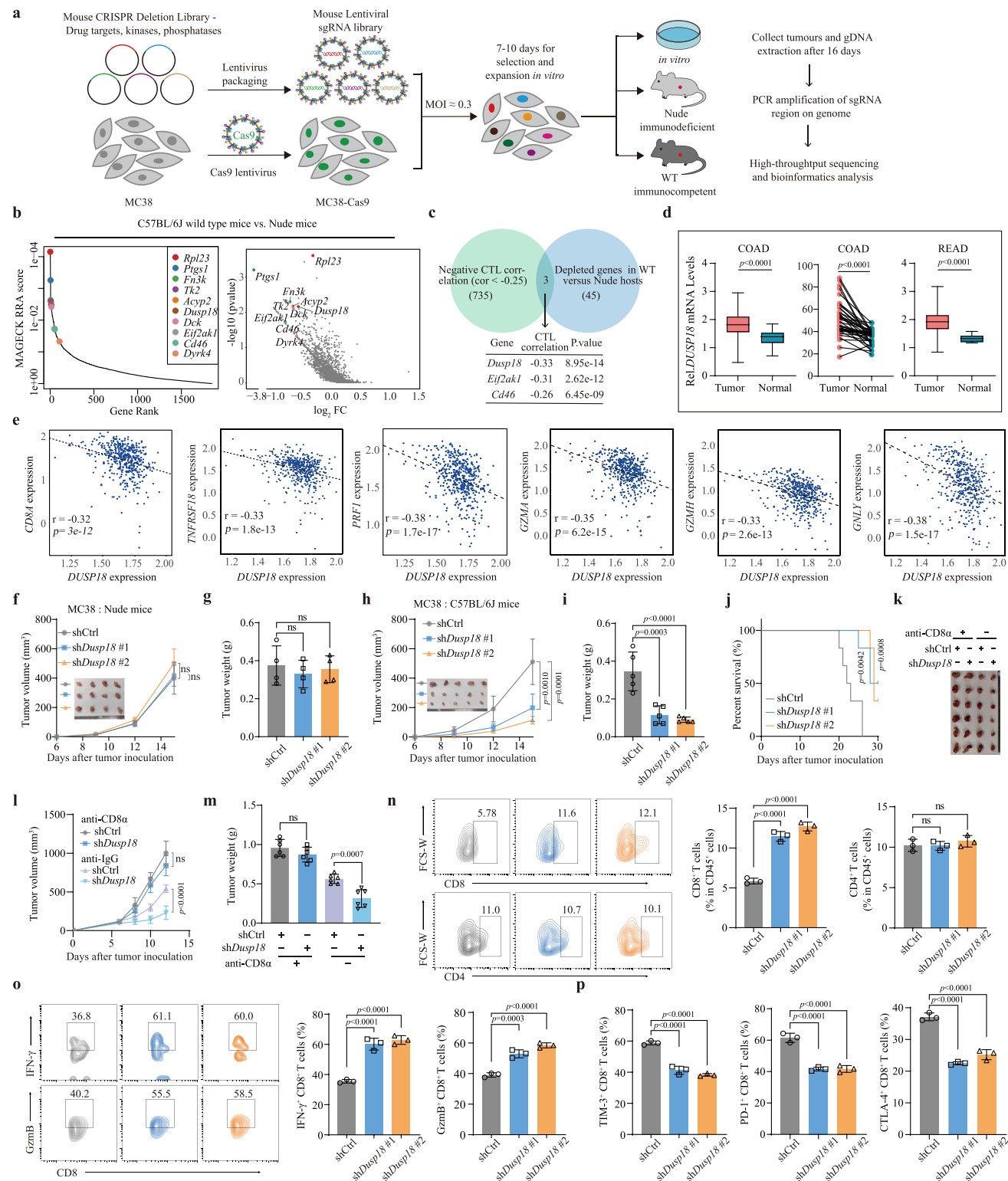

tumor growth in immunodeficient nude mice (Fig. 1f, g and Supplementary Fig. 1j). Similar findings were observed with the human HCT116 CRC cell line (Supplementary Fig. 1k–m). However, *Dusp18* inhibition did impair tumor growth and prolonged survival in immunocompetent mice (Fig. 1h-j) although depleting CD8+ T cells completely eliminated this growth disadvantage (Fig. 1k–m and Supplementary Fig. 1n). Thus, our findings suggested that CD8+ T-cell-mediated immunity is involved in some aspect(s) of DUSP18-regulated tumor growth.

The effect of *Dusp18* inhibition on the tumor immune landscape was further explored using flow cytometry. This showed that the absolute numbers and percentage of innate cells in shCtrl and sh*Dusp18* tumors were identical (Supplementary Fig. 1o, p). Although both CD8+ and CD4+ T cell numbers were increased in sh*Dusp18* tumors, the proportion of these immune cells was enhanced only by CD8+ but not CD4+ T cells (Fig. 1n and Supplementary Fig. 1p). Immunohistochemical staining also showed that inhibition of *Dusp18* significantly upregulated CD8+ T cell infiltration with little effect on CD4+

**Fig. 1 | Inhibition of Dusp18 enhances tumor-infiltrating CD8⁺ T cell function.**
**a** Workflow of in vivo screens to identify potential targets for immune evasion.
**b** MAGeCK analysis and RRA ranking of top depleted genes in the screens (n = 2).
Ranked dot plots of depleted genes in immunocompetent hosts compared with
immunodeficient hosts are shown. **c** Venn diagram showing the overlap of
45 screened genes and 735 CTL related genes in the TCGA-COAD database.
**d** Differential expression analysis for *DUSP18* in tumors and normal tissues. COAD,
colon adenocarcinoma (tumor samples (n = 471), normal samples (n = 41) (left);
tumor samples (n = 41), paired normal samples (n = 41) (middle)). READ, rectal
adenocarcinoma (tumor samples (n = 167), normal samples (n = 10) (right).
**e** Scatterplot showing the correlation of the *DUSP18* expression with that of several
activated T cell-related genes in TCGA-COAD samples (n = 471 tumor samples).
Tumor growth curves (**f**) or tumor weight (**g**) for 4–6 weeks nude mice bearing
MC38 tumors (n = 4). Tumor growth curves (**h**), tumor weight (**i**) or survival curves

for 4–6 weeks old C57BL/6 bearing MC38 tumors (n = 5). Tumor photo (**k**), tumor
growth curves (**l**) and tumor weight (**m**) for C57BL/6 bearing MC38 tumors (n = 6).
Mice were treated with anti-CD8a antibodies on days −1, 0, 7, and 14, **n** FACS ana-
lysis of tumor-infiltrating CD4⁺ and CD8⁺ T cells in shCtrl or *shDusp18* MC38 tumors.
(n = 3). **o** Quantification of IFN-γ and GzmB CD8⁺ TILs. Tumor-infiltrating T cells
were pre-stimulated with PMA, ionomycin and brefeldin A for 3 h (n = 3). IFN-γ and
GzmB-producing cells were determined by flow cytometry. **p** Quantification of PD-
1, TIM-3 and CTLA-4 CD8⁺ TILs through flow cytometry (n = 3). Data are presented
as mean ± SD (**f**–**p**). *P* values were calculated by MAGeCK-test module using a
modified robust ranking aggregation (α-RRA) analysis (**b**), unpaired two-tailed t-
tests (**d:** left and right boxplot, **l, m**), paired two-tailed t-tests (**d** middle boxplot),
one-way ANOVA (**f, g, h, i, n**–**p**) or log-rank (Mantel-Cox) test (**j**). *P* values and R were
calculated by Spearman's correlation analysis. Two-sided *P* value was given (**c, e**); ns
not significant. Source data are provided as a Source Data file.

---

T cells and NK cells (Supplementary Fig. 2a). Infiltrating CD8⁺ T cells
from sh*Dusp18* tumors showed elevated expression of cytotoxic
molecules IFN-γ and granzyme B (Fig. 1o), alongside reduced expres-
sion of exhaustion-associated transcripts PD-1, TIM-3, and CTLA-4
(Fig. 1p and Supplementary Fig. 2b). After MC38 cells overexpressing
wild-type (WT) or phosphatase dead mutant (Dead) *Dusp18* were
inoculated into mice, only the former promoted tumor growth in
immunocompetent mice but not in nude mice (Supplementary
Fig. 2c, d). Additionally, only WT *Dusp18* overexpression inhibited
CD8⁺ T cell infiltration (Supplementary Fig. 2e, f) and cytotoxic func-
tion (Supplementary Fig. 2g). The expression of exhaustion-associated
molecules in CD8⁺ T were also higher in WT *Dusp18* overexpressing
tumors (Supplementary Fig. 2h). Thus, the above results showed that
DUSP18's inhibition of CD8⁺ T cell infiltration and its associated cyto-
toxicity is dependent on its phosphatase activity.

## DUSP18 inhibition impairs cholesterol biosynthesis
To determine whether the attenuation of DUSP18 expression alters
intestinal tumorigenesis, we crossed *Dusp18*^flox/flox^ mice with *Villin*^cre^
mice to generate mice with intestinal epithelium-specific depletion of
*Dusp18* (termed CKO mice) (Supplementary Fig. 2i). Wild-type (WT)
and CKO mice were then used to generate CRCs using the AOM/DSS
protocol (Fig. 2a). Surprisingly, CKO mice developed significantly
fewer CRCs (Fig. 2b) and survived longer (Fig. 2c). To demonstrate that
*Dusp18* loss was triggering an adaptive immune response, WT and CKO
tumors were examined for evidence of infiltration by CD8⁺ T cells,
CD4⁺ T cells and NK cells. Immunofluorescence studies showed WT
tumors to be immune deserts that were largely devoid of CD8⁺ T cells
(Fig. 2d) whereas the isogenic CKO tumors were highly infiltrated by
CD8⁺ T cells (Fig. 2d). In contrast, CD4⁺ T cell and NK cell infiltration did
not change significantly (Supplementary Fig. 2j). Expression of the cell
proliferation markers Ki-67 also did not differ in the two tumor tissues
(Supplementary Fig. 2j). Tumors from CKO mice had slightly higher
levels of JNK phosphorylation (Phospho-JNK (Tyr185)), while ERK and
p38 remained almost unchanged among several MAPKs (Supplemen-
tary Fig. 2k). The expression of c-Myc, Cyclin D1, and Cox-2 was not
significantly different in these two tumor tissues (Supplemen-
tary Fig. 2k).

To establish the molecular mechanism(s) by which DUSP18 sup-
presses antitumor immunity, RNA-seq was performed using shCtrl and
sh*Dusp18* MC38 cells. Gene ontology (GO) and Kyoto Encyclopedia of
Genes and Genomes (KEGG) analysis showed that the differentially
expressed genes were involved in cholesterol biosynthesis and its
downstream metabolism and were expressed at lower levels in
sh*Dusp18* cells (Fig. 2e–g). Gene set enrichment analysis (GSEA) in
these cells also showed that genes involved in cholesterol homeostasis
were positively enriched in the control cells (Fig. 2h). We further
confirmed decreased mRNA expression and protein levels for selected
genes in sh*Dusp18* MC38 cells and CKO tumor tissues (Fig. 2i-l and
Supplementary Fig. 3a). RNA-seq was also performed in shCtrl and

sh*DUSP18* human CRC HCT116 cells to validate these results from mice
MC38 cells. Both GO, KEGG and GSEA analyses revealed that inhibition
of *DUSP18* significantly reduced the expression of genes involved in
the cholesterol biosynthesis pathway (Supplementary Fig. 3b–f). We
confirmed decreased mRNA expression and protein levels for selected
cholesterol biosynthesis genes in sh*DUSP18* HCT116, SW480 and
additional human CRC cells (Supplementary Fig. 3g–j). Proteomic
analysis in MC38 cells also validated the above results (Supplementary
Data 4). As shown in Supplementary Fig. 4a, inhibition of *Dusp18*
reduced the levels of 440 proteins and increased the levels of 449
proteins. KEGG signaling pathway enrichment analysis of these dif-
ferentially expressed proteins showed that those which were down-
regulated in sh*Dusp18* cells were mainly enriched in metabolic path-
ways and cholesterol biosynthesis signals (Supplementary Fig. 4b),
whereas upregulated proteins were mainly enriched in antigen pre-
sentation signals (Supplementary Fig. 4c). These results were con-
sistent with our RNA-Seq results. Of particular note, inhibition of
*Dusp18* decreased the levels of SREBP2, HMGCR, LSS, and SQLE
(Fig. 2m). Collectively above results underscored the importance and
conservation of DUSP18 in regulating the cholesterol synthesis
pathway.

*DUSP18* expression levels were positively correlated with choles-
terol biosynthesis signaling in TCGA-COAD and GEO datasets (Fig. 2n
and Supplementary Fig. 4d). Since DUSP18 appeared to act as a
phosphatase to regulate the cholesterol synthesis signaling pathway,
we hypothesized that DUSP18 regulates a TF that is involved in main-
taining the pathway. We examined the expression of several important
TFs involved in the cholesterol synthesis pathway based on the inhi-
bition of *Dusp18*, including SREBP2, SREBP1, Ad4BP, Maf, Notch1, XBP1,
RORγ, and c-Myc[28,36,44–49]. As shown in Supplementary Fig. 4e, inhibi-
tion of *Dusp18* only down-regulated SREBP2 protein levels and had
little effect on the protein levels of the other TFs. Decreased expres-
sion of *SREBF2* mRNA and SREBP2 protein levels were validated in
human and murine CRC cells expressing *DUSP18* shRNA (Fig. 2i-l and
Supplementary Fig. 3a, g–j). Importantly, levels of both the full (p-
SREBP2) and cleaved (n-SREBP2) forms of SREBP2 protein[28] were
decreased. Additionally, *DUSP18* and *SREBF2* mRNA levels were sig-
nificantly positively correlated in TCGA-COAD (Supplementary Fig. 4f),
and pan-cancer analyses revealed a similar correlation in most tumors,
including CRCs, liver and kidney cancers (Supplementary Fig. 4g).
Overall, these findings support the notion that DUSP18 exerts control
over the tumor-associated immune landscape by regulating cellular
cholesterol biosynthesis.

## USF1 is essential for DUSP18-mediated regulation of cholesterol biosynthesis
In order to elucidate how *DUSP18* regulates the expression of SREBF2
in CRC, we asked whether DUSP18 interacts with a TF that regulates
*SREBF2* gene expression. GSEA of our RNA-seq data revealed *USF1*
transcripts to be most significantly enriched, thus leading us to

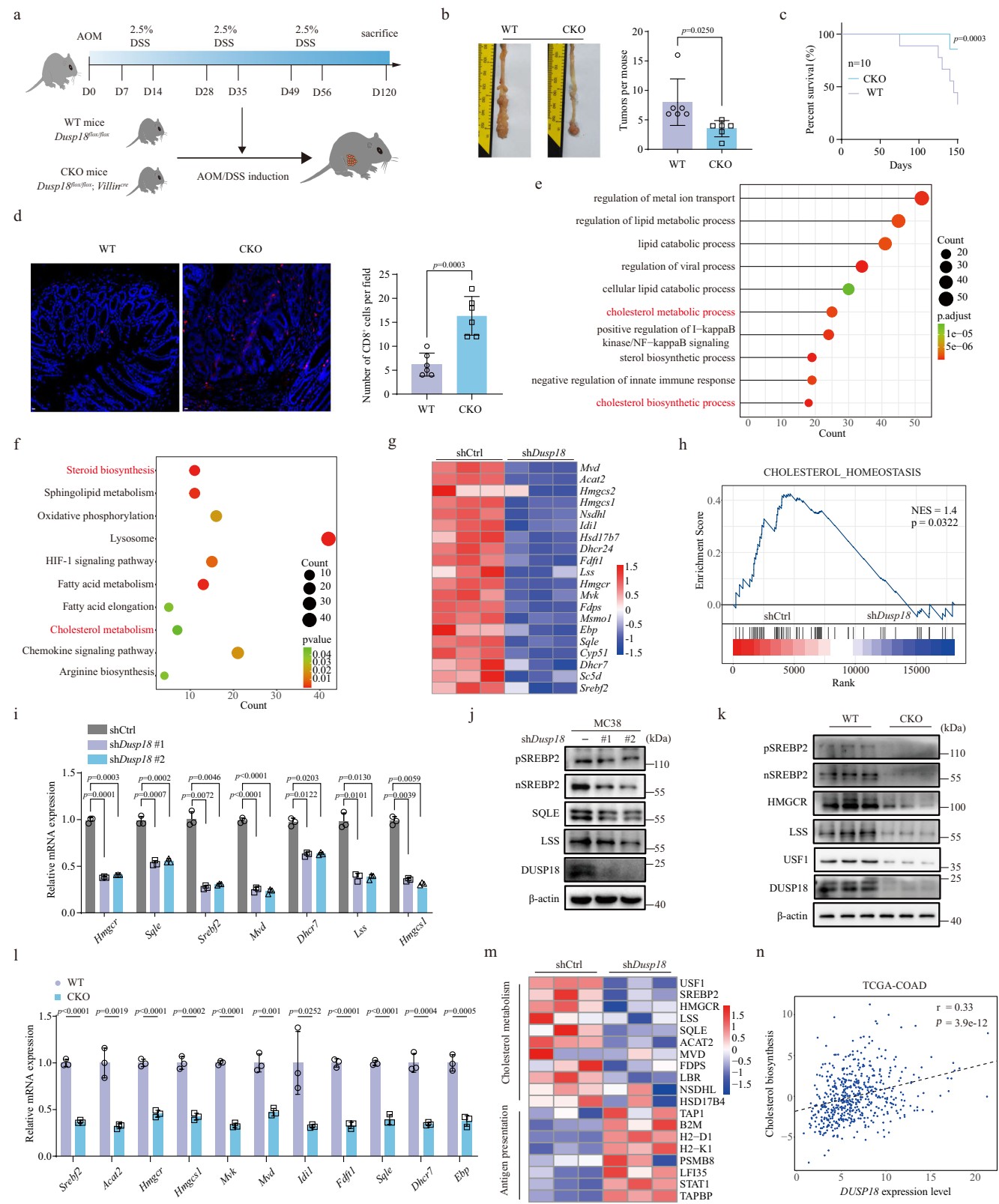

hypothesize that USF1 may be the TF that indirectly regulates cholesterol metabolism perhaps via an association with SREBP2 (Fig. 3a). Using the Human TFDB and JASPAR websites to predict the putative binding sites for USF1 in the *SREBF2* promoter (Supplementary Fig. 5a), and analyzing of ChIP-seq data from HCT116 cells showed a USF1-binding peak in the *SREBF2* promoter (Supplementary Fig. 5b). The functional consequences of this were established by demonstrating

that inhibition of *USF1* significantly down-regulated both *SREBF2* mRNA and SREBP2 protein levels (Fig. 3b and Supplementary Fig. 5c). ChIP assays also revealed that USF1 occupied the *SREBF2* gene promoter in HCT116 cells (Fig. 3c). To confirm this, luciferase reporters containing *SREBF2* promoter elements were rendered USF1-responsive, whereas inhibition of *USF1* dramatically decreased luciferase activity driven by a WT *SREBF2* promoter, but not by a promoter that

**Fig. 2 | Dusp18 inhibition decreases cholesterol biosynthesis. a** Scheme for the AOM/DSS-induced colon cancer model in WT and CKO mice. **b** Colon tumor numbers in mice from (**a**), $n = 6$. **c** Survival curves, $n = 10$. **d** Representative IF images and quantification of CD8$^+$ T cells in AOM/DSS-induced colon cancer formed by WT and CKO mice. Scale bar, 10 μm. For each mouse, five fields from one independent tumor of each mouse were analyzed ($n = 6$). GO (**e**) and KEGG (**f**) analyses show the altered pathways after *Dusp18* inhibition. **g** Heatmap of RNA-seq expression values of cholesterol biosynthesis genes based on sh*Dusp18* and shCtrl MC38 cells ($n = 3$). **h** GSEA analysis for cholesterol homeostasis pathway genes in shCtrl versus sh*Dusp18* cells. **i** qRT-PCR analysis of the indicated cholesterol biosynthetic pathway genes in sh*Dusp18* and shCtrl MC38 cells ($n = 3$). **j** Western blot analysis for the indicated proteins in sh*Dusp18* and shCtrl MC38 cells. The IB data are representative of three independent experiments. **k** Immunoblot (IB) analysis of the indicated proteins in CRCs from AOM/DSS-treated mice, $n = 3$ independent experiments. **l** qRT-PCR analysis of the indicated cholesterol biosynthetic pathway genes in CRCs from WT and CKO mice, $n = 3$ independent experiments. **m** Heatmap showing differential expression of genes in the cholesterol metabolism and antigen presentation in the proteomics of shCtrl and sh*Dusp18* MC38 cells ($n = 3$). **n** Scatterplot showing the correlation between *DUSP18* expression levels with those of cholesterol biosynthesis-related genes in TCGA-COAD samples ($n = 473$). Spearman correlation coefficient ($r$) and $p$ value are marked. Data are presented as mean ± SD (**b, d, i, l**). $P$ values were calculated by modified Fisher's exact tests (**e, f**), Kolmogorov–Smirnov tests (**h**), unpaired two-tailed $t$ tests (**b, d, l**), one-way ANOVA (**i**) or log-rank (Mantel-Cox) test (**c**). $P$ values and R were calculated by Spearman's correlation analysis. Two-sided $P$ value was given (**n**); ns not significant. Source data are provided as a Source Data file.

---

was unable to bind USF1 (Fig. 3d). These data indicated that USF1 can bind to the *SREBF2* promoter and regulate its transcriptional output. In GSEA analysis, inhibition of *DUSP18* led to a notable down-regulation of the USF signature (Fig. 3e).

Based on the above conjecture, we next asked whether DUSP18 interacts with USF1 so as to affect the expression of *SREBF2*. Coimmunoprecipitation assays demonstrated interactions between both endogenous and exogenous DUSP18 and USF1 (Supplementary Fig. 5d, e). Deletion analyses showed that the N terminal domain of USF1 and the DSPc domain of DUSP18 were crucial for this interaction (Supplementary Fig. 5f, g). However, inhibition of *DUSP18* decreased USF1 protein without affecting its mRNA level (Fig. 3f and Supplementary Fig. 5h). Consistent with this, inhibition of *DUSP18* significantly reduced the half-life of endogenous USF1 (Supplementary Fig. 5i).

Since the previous results indicated that DUSP18's ability to block T cell activation requires its phosphatase activity (Supplementary Fig. 2c–h), its ability to dephosphorylate and activate USF1 was investigated. To this end, Flag-tagged WT and enzymatically dead *DUSP18* and HA-tagged *USF1* were co-expressed in HEK293T cells followed by anti-HA antibody and immunoblotting with anti-pan phospho-Tyr, Ser and Thr antibodies. This showed that DUSP18 selectively dephosphorylates USF1 at one or more Thr residues and suggested that this is responsible for stabilizing USF1 and allowing it to transcriptionally activate *SREBF2* (Fig. 3g). After serially mutating individual Thr residues in *USF1* and co-expressing these with HA-*DUSP18* in HEK293T cells, we found that the USF1 mutant T100A no longer served as a dephosphorylation substrate for HA-DUSP18 (Fig. 3h). Consistent with the above hypothesis, cycloheximide chase experiments indicated that USF1$^{T100A}$ had an extended half-life relative to that of either WT USF1 or the phospho-mimetic USF1$^{T100D}$ mutant (Fig. 3i). USF1$^{T100A}$ was also less subject to ubiquitination (Fig. 3j), as well as having more binding with USF2 (Supplementary Fig. 5j), whose binding is important for USF1 to perform its transcriptional function as previously noted[50]. In contrast to this, binding to the transcriptional repressor Cha was not significantly altered (Supplementary Fig. 5k).

Casein kinase 2 (CK2) is a known kinase for USF1 T100[50]. To examine the cross-talk between DUSP18 and CK2 on USF1 threonine phosphorylation, DUSP18 and CK2β were co-expressed or co-depleted in CRC cells. The studies showed that DUSP18 decreased CK2β-mediated phosphorylation of USF1 (Supplementary Fig. 5l, m). Therefore, DUSP18 and CK2β appear to function antagonistically to regulate USF1 stability.

Finally, we found that, whereas *DUSP18* overexpression did not upregulates the SREBP2 in CRC cells with inhibition of *USF1* (Fig. 3k), *USF1* overexpression could largely rescue the SREBP2 protein level decline mediated by *DUSP18* inhibition (Supplementary Fig. 5n). These results suggest that USF1 plays an indispensable role in the regulation of SREBP2 by DUSP18. In addition, overexpression of *USF1*$^{T100A}$ but not *USF1*$^{T100D}$ rescued the inhibition of CRC cell growth mediated by *Dusp18* inhibition (Fig. 3l). *USF1*$^{T100A}$ overexpressing tumors exhibited

fewer CD8$^+$ T cells infiltration (Fig. 3m) and lower expression levels of the cytotoxic molecules IFN-γ and granzyme B (Fig. 3n). In contrast, they expressed higher levels of T cell exhaustion molecules such as PD-1, TIM-3 and CTLA-4 (Fig. 3o). Together, these findings point to the existence of a transcriptional cascade in which DUSP18 dephosphorylates and stabilizes USF1, which in turn upregulates SREBP2 and cholesterol biosynthetic signaling.

## DUSP18 ablation induces T cell activation in vitro

Because CD8$^+$ T cells must recognize MHC-I molecules on the surface of tumor cells prior to initiating tumor cell killing (MHC restriction), the effect of *Dusp18* inhibition on tumor cell antigen presentation was examined. Two well-established mouse synergic tumor models, MC38-OVA and B16-OVA, were selected. A higher level of MHC-I (H2-Kb) was detected in MC38-OVA and B16-OVA cells with *Dusp18* inhibition (Fig. 4a, c), as well as increased MHC-I-bound SIINFEKL (OVA epitope peptide) complex expression (Fig. 4b, d). Since *Dusp18* mediates tumor immune evasion in a CD8$^+$ T cell-dependent manner, the cytotoxic killing of B16-OVA-Luc and MC38-OVA-Luc cells (expressing Ovalbumin and Luciferase) by OT-I T cells was measured in an in vitro killing system (Fig. 4e). This demonstrated that apoptosis and cell death were more pronounced in sh*Dusp18* cells (Fig. 4f, g). We next co-cultured tumor cells and CD8$^+$ T cells to explore the effect of *Dusp18* inhibition on CD8$^+$ T cell activation, effector function, and expression of exhausted-related molecules (Fig. 4h). Inhibition of *Dusp18* significantly enhanced expression of activation molecules CD69 and CD25, IFN-γ and GzmB production of co-cultured CD8$^+$ T cells (Fig. 4i, j and Supplementary Fig. 6a). In addition, the percentage of PD-1$^+$, TIM-3$^+$, CTLA-4$^+$ and PD-1$^+$TIM-3$^+$ CD8$^+$ T cells decreased in the sh*Dusp18* cohort (Fig. 4k, l and Supplementary Fig. 6b, c). Collectively, these findings demonstrated that DUSP18 in tumor cells suppresses CD8$^+$ T cell activation and cytotoxicity and promotes CD8$^+$ T cell exhaustion.

## Tumor-cell-derived lanosterol in the TME diminishes CD8$^+$ T cell activation by inhibiting KRAS-ERK signaling

Because the preceding work showed that DUSP18 and USF1 collaborate to upregulate cholesterol biosynthesis, we explored the possibility that one or more intermediates in the cholesterol biosynthesis pathway might be responsible for suppressing CD8$^+$ T cell function. We therefore determined the levels of these intermediates in tumor interstitial fluid of sh*Dusp18* and shCtrl MC38 tumors using cholesterol metabolomics. The levels of cholesterol itself, numerous cholesterol synthesis intermediates, oxysterols and other derivatives were significantly lower in the fluid obtained from sh*Dusp18* tumors (Fig. 5a and Supplementary Data 5). The most down-regulated of these (40%) was lanosterol, a little-studied cholesterol synthesis intermediate (Fig. 5b, c). To determine the relevance of this directly, primary CD8$^+$ T cells were treated with different concentrations of lanosterol and the result showed that lanosterol reduced expression of the CD8$^+$ T cell activation markers CD69 in a dose dependent manner (Fig. 5e). In another set of experiments, we added Ro 48-8071, an inhibitor of

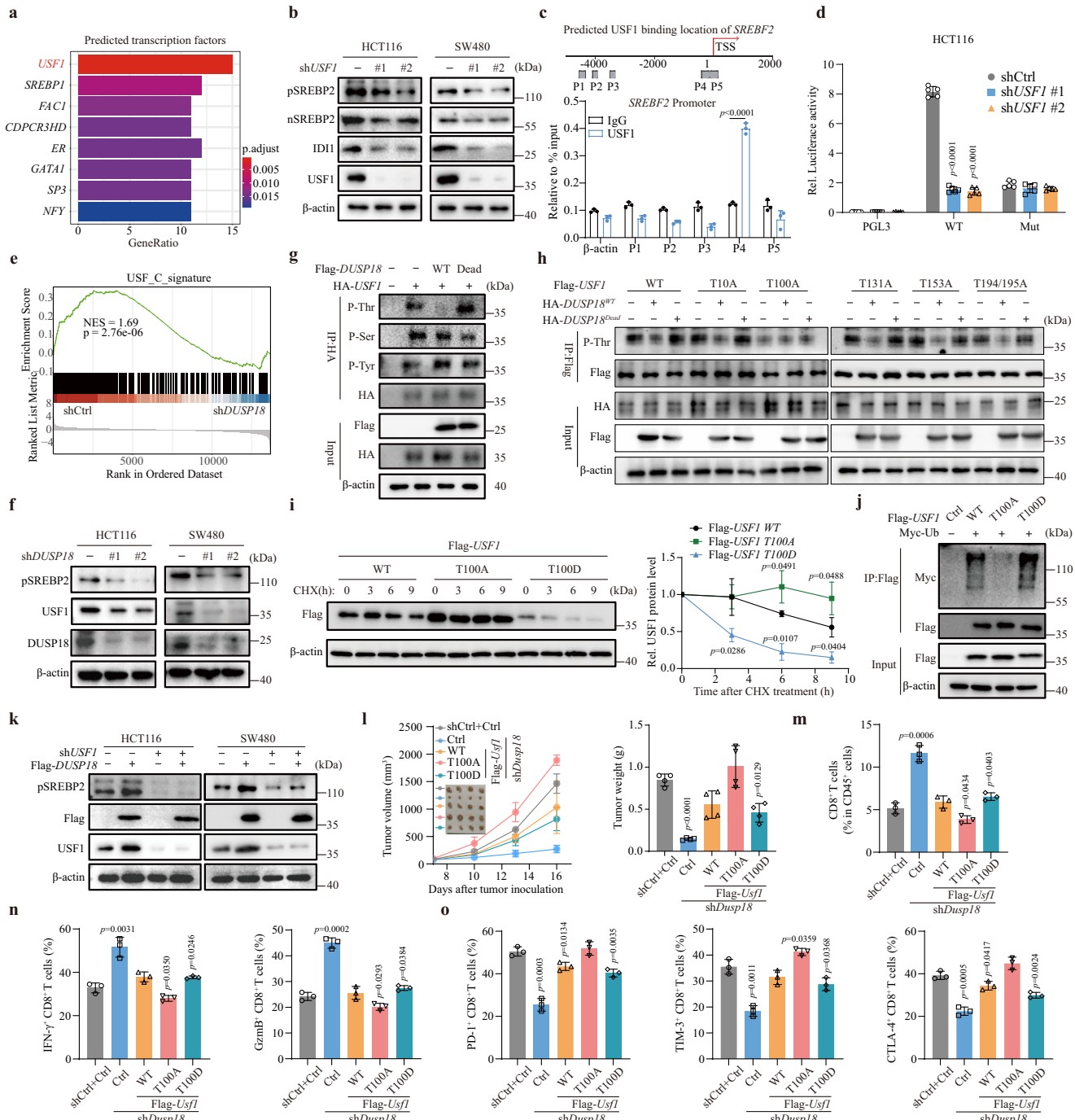

**Fig. 3 | USF1 is a downstream target of DUSP18. a** Candidate transcription factor prediction was performed by GSEA of regulatory target gene sets. The top 8 TFs ranked by GeneRation scores and p values are shown. **b** Inhibition *of USF1* suppresses the indicated protein levels. **c** qPCR ChIP analysis of USF1 binding to *SREBF2* promoter regions in HCT116 cells (*n* = 3). **d** A luciferase vector driven by a WT *SREBF2* promoter, but not by a non-USF1-binding mutant promoter, is responsive to inhibition of *USF1* (*n* = 3). **e** GSEA analysis for USF signature in shCtrl versus sh*DUSP18* cells. **f** *DUSP18* inhibition suppresses USF1 protein levels. **g** USF1 threonine dephosphorylation mediated by DUSP18. HA-*USF1* was co-transfected with Flag-*DUSP18* (WT or phosphorylase Dead) into HEK293T cells, and the cell lysates were subjected to immunoprecipitation. **h** Dephosphorylation of USF1 T100 by DUSP18. Flag-*USF1s* (WT, T10A, T100A, T131A, T153A and T194/195A) were co-transfected with HA-*DUSP18*^WT or *DUSP18*^Dead into HEK293T cells, and the cell lysates were subjected to immunoprecipitation. **i** Stability of Flag-USF1 (WT, T100A and

T100D) proteins in HEK293T cells (left) in the presence of CHX block. Flag-USF1 protein quantified by densitometry, with β-actin as a loading control (right) (*n* = 3). **j** Decreased Flag-USF1 (T100A) polyubiquitination. Myc-Ub was co-transfected with Flag-USF1 (WT, T100A and T100D) into HEK293T cells, and the cell lysates were subjected to immunoprecipitation. **k** Overexpression of Flag-*DUSP18* has no effect on the SREBP2 protein level mediated by *USF1* inhibition. **l** Subcutaneous xenograft experiments in C57BL/6 were performed in the indicated MC38 cells group. Tumor growth curves and weights are shown (*n* = 4). The percentage of infiltrating CD8^+ T cells (*n* = 3) (**m**), effector molecules (*n* = 3) (**n**), and exhausted molecules (*n* = 3) (**o**) in the indicated group. Data are presented as mean ± SD (**c**, **d**, **i**, **l–o**). *P* values were calculated by modified Fisher's exact tests (**a**), Kolmogorov–Smirnov tests (**e**), unpaired two-tailed t-tests (**c**, **i**), one-way ANOVA (**d**, **l–o**); ns, not significant. All IB data are representative of three independent experiments. Source data are provided as a Source Data file.

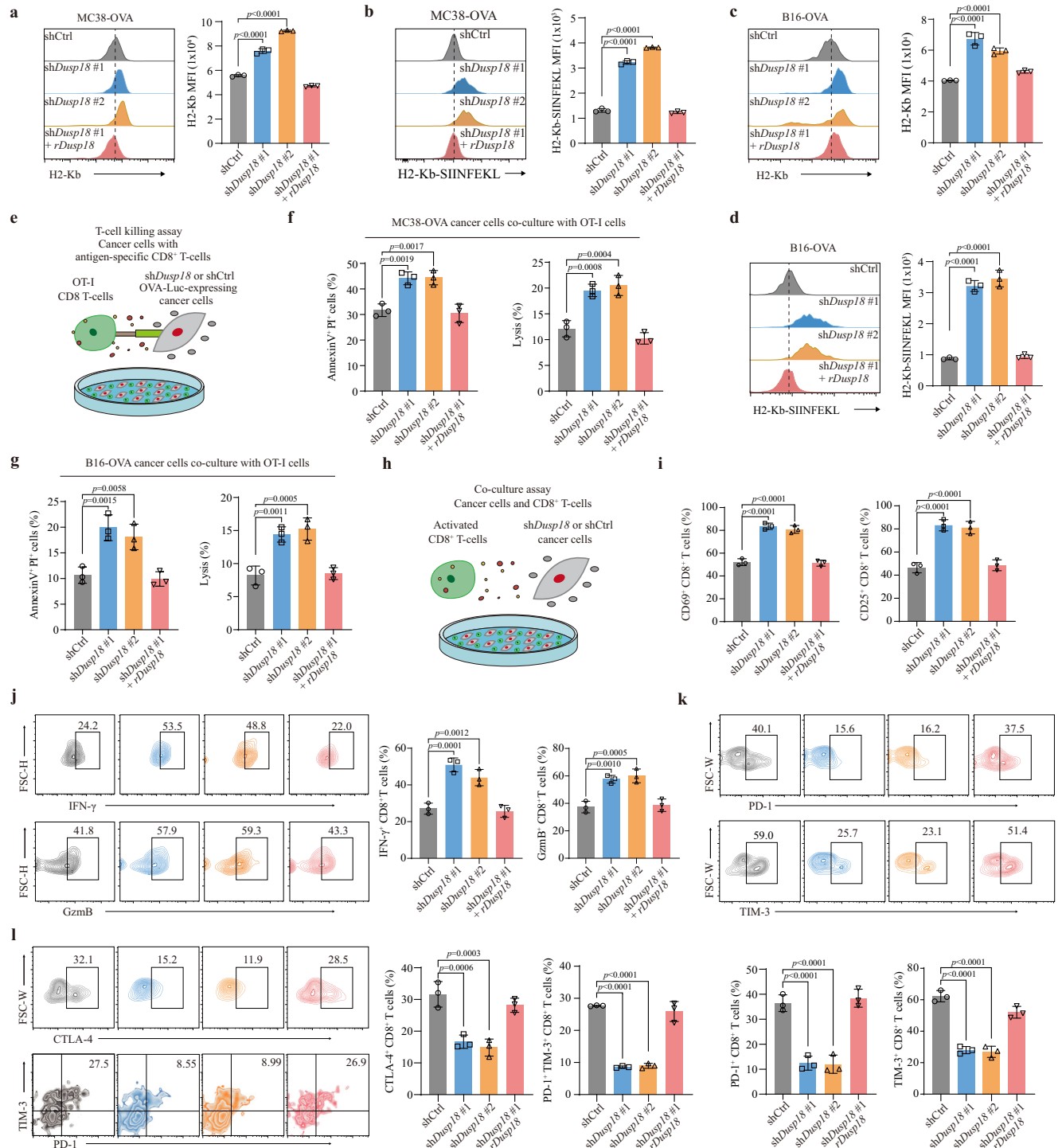

**Fig. 4 | DUSP18 ablation induces T cell activation in vitro.** Expression levels of H2-Kb and H2-Kb-SIINFEKL on sh*Dusp18* and shCtrl MC38-OVA ($n = 3$) (**a**, **b**) and B16-OVA ($n = 3$) (**c**, **d**) cells were determined by FACS. (MFI, mean fluorescence intensity). **e** Schematic representation of the in vitro T cell killing assay. MC38-OVA-Luc or B16-OVA-Luc shCtrl and sh*Dusp18* cells were co-cultured with splenic CD8+ T cells from OVA-specific T cell receptor transgenic (OT-I) mice. Cytotoxic effects of OT-I were measured by Annexin V/propidium iodide staining ($n = 3$) and biolumi-nescence signaling ($n = 3$) from MC38-OVA (**f**) and B16-OVA (**g**) cells after being co-cultured with OT-I for 48 h. **h** Schematic representation of the ex vivo T cell co-culture assay. Splenic CD8+ T cells were activated with the CD3 and CD28

antibodies, and co-cultured with sh*Dusp18* or shCtrl tumor cells. After 24 h of coculture, cells were harvested and processed for flow cytometry analysis. **i** Quantification of CD69 (left) or CD25 (right) expression in CD8+ T cells after co-culture with MC38 cells. ($n = 3$). **j** Quantification of IFN-γ (left) or GzmB (right) production in CD8+ T cells after co-culture with MC38 cells ($n = 3$). **k, l** Quantification of PD-1+, TIM-3+, CTLA-4+ and PD-1+ TIM-3+ percentage in CD8+ T cells after co-culture with MC38 cells ($n = 3$). Data are presented as mean ± SD (**a**–**d**, **f**, **g**, **i**–**l**). *P* values were calculated by one-way ANOVA; ns, not significant. Source data are provided as a Source Data file.

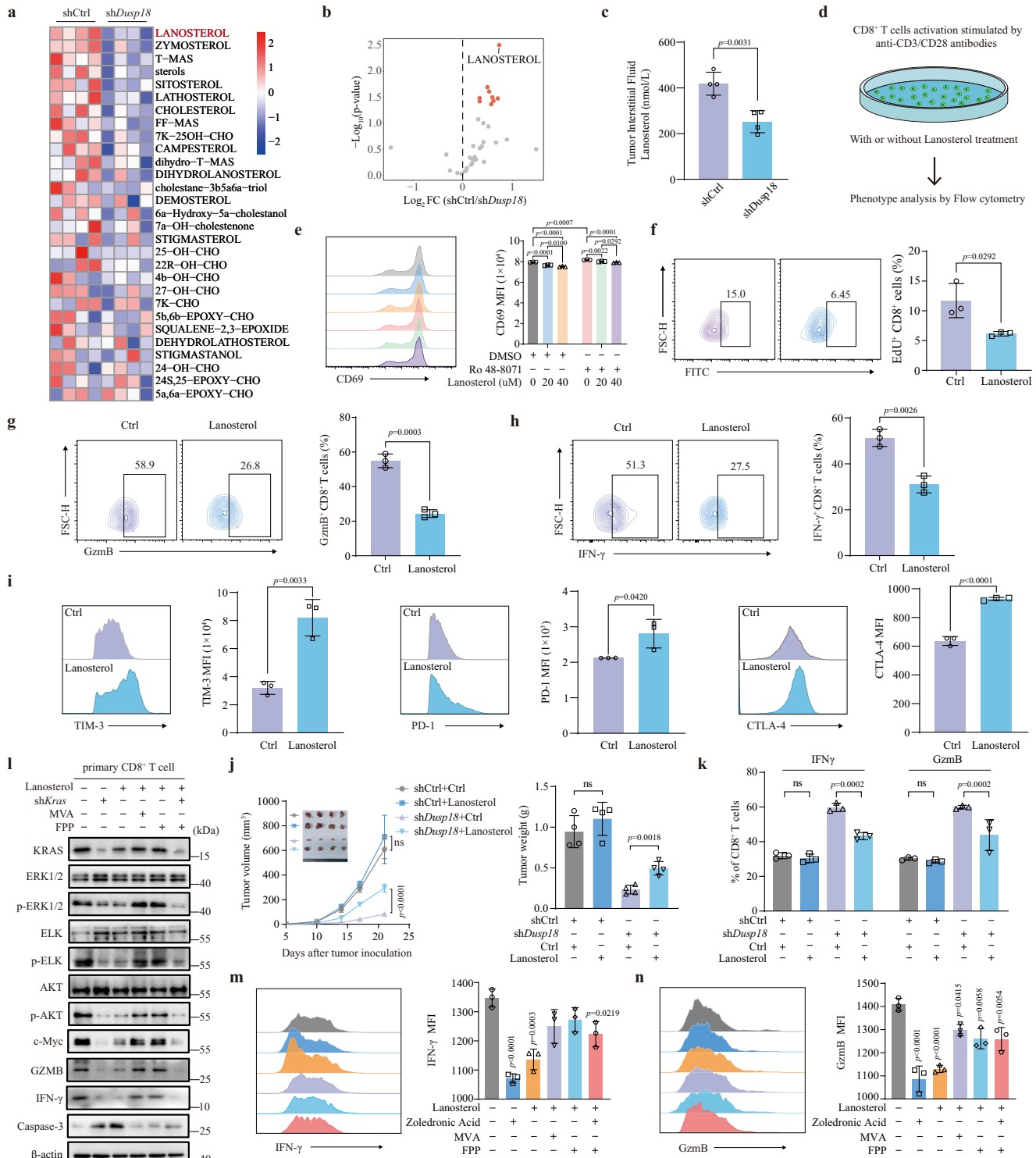

**Fig. 5 | Tumor-cell-derived lanosterol in the TME promotes CD8+ T cell inactivation. a** Cholesterol metabolism-related metabolites and absolute lanosterol concentrations in interstitial fluids from subcutaneous sh*Dusp18* MC38 tumors in 4–6 weeks old C57BL/6J mice (n = 4). **b** Volcano plot showing differences in lanosterol and other metabolites in shCtrl and sh*Dusp18* tumors. **c** Boxplot of absolute lanosterol concentrations in tumor interstitial fluid (n = 4). **d** Schematic representation of CD8+ T cells phenotypes treated by lanosterol. **e** FACS analysis of CD8+ T cell activation marker CD69 with indicated treatment (n = 3). **f** FACS analysis of CD8+ T cell proliferation with lanosterol treatment. Quantification of IFN-γ (n = 3) (**g**) and GzmB (n = 3) (**h**) production in CD8+ T cells after lanosterol treatment. **i** Quantification of PD-1+ (n = 3), TIM-3+ (n = 3) and CTLA-4+ (n = 3) MFI in CD8+ T cells

after lanosterol treatment. 1 × 10⁶ shCtrl or sh*Dusp18* MC38 cells were subcutaneously injected into C57BL/6J mice (n = 4). Mice were injected with 30 mg/kg of lanosterol every 2 days from day 0 until the time of sacrificed. Tumor growth curves and photos of tumors are shown in (**j**); Flow cytometric quantification of IFN-γ (left) and GzmB (right) CD8+ T cells from different groups is shown in (**k**) (n = 3). **l** The indicated protein levels were confirmed by Western blotting following the indicated treatments. The IB data are representative of three independent experiments. Quantification of IFN-γ (**m**) and GzmB (**n**) MFI in CD8+ T cells following the indicated treatment (n = 3). Data are presented as mean ± SD (**c**, **e–i**, **j**, **k**, **m**, **n**). P values were calculated by unpaired two-tailed t tests (**b**, **c**, **f–i**, **j**, **k**), one-way ANOVA (**e**, **m**, **n**); ns not significant. Source data are provided as a Source Data file.

lanosterol synthase (LSS) to block endogenous lanosterol production. Ro 48-8071 treatment enhanced CD69 expression in CD8[+] T cells, and a gradient of lanosterol was able to decrease CD69 expression when the CD8[+] T cells with or without the addition of Ro 48-8071 (Fig. 5e). Lanosterol treatment not only led to a decrease in CD25 expression (Supplementary Fig. 7a) but also effectively inhibited CD8[+] T cell proliferation (Fig. 5f and Supplementary Fig. 7b). Furthermore, it notably suppressed cytokine production, including IFN-γ and GzmB (Fig. 5g, h), while concurrently promoting the expression of immune checkpoint molecules such as PD-1, TIM-3, and CTLA-4 (Fig. 5i and Supplementary Fig. 7c), as well as inducing cellular apoptosis (Supplementary Fig. 7d).

Relative to control CD8[+] T cells, those treated with lanosterol showed impaired ability in their killing of MC38-OVA and B16-OVA tumor cells. In addition, the proportion of apoptotic tumor cells was significantly reduced in the lanosterol-treated group, as was LDH release and the number of viable tumor cells was significantly increased (Supplementary Fig. 7e). To assess the impact of lanosterol on in vivo tumor growth, we subcutaneously inoculated equal numbers of shCtrl and sh*Dusp18* MC38 cells. Notably, silencing *Dusp18* resulted in a deceleration of xenograft tumor growth (Fig. 5j), accompanied by increased levels of IFN-γ and GzmB expression (Fig. 5k), and a decrease in the percentage of PD-1[+], TIM-3[+], and CTLA-4[+] CD8[+] T cells (Supplementary Fig. 7f). Intriguingly, this effect was partially counteracted by intraperitoneal (i.p.) injection of lanosterol (30 mg/kg mice) (Fig. 5j, k and Supplementary Fig. 7f).

Lanosterol has been reported to promote ubiquitination and proteasomal degradation of HMG-CoA reductase (HMGCR), the rate-limiting enzyme in the mevalonate pathway[51]. Consistent with this, lanosterol reduced HMGCR protein levels in a dose-dependent manner both in mouse primary CD8[+] T cells and in human Jurkat T lymphoblastic leukemia cells (Supplementary Fig. 7g).

The mevalonate pathway is essential for the synthesis of a variety of isoprenoids derived from acetyl-CoA, including prenyl groups, which are essential for the in vivo biological activity of RAS proteins[30]. Lanosterol treatment significantly inhibited KRAS protein abundance as well as ERK, and AKT signaling in mouse primary CD8[+] T cells, while activating apoptosis-associated caspase-3 cleavage as previously reported (Fig. 5l)[52]. Zoledronic Acid, an inhibitor of farnesyl pyrophosphate synthase, blocked the synthesis of farnesyl pyrophosphate (FPP) and geranylgeranyl pyrophosphate while reducing KRAS prenylation and attenuating downstream signaling. Like lanosterol, zoledronic acid also inhibited IFN-γ and GzmB production by CD8[+] T cells. Supplementation with either mevalonic acid or FPP largely reversed the signaling inhibition mediated by both (Fig. 5m, n).

KRAS signaling-mediated T cell receptor (TCR) signaling activation and cell proliferation are prerequisites for CD8[+] T function, which may explain why lanosterol inhibits intratumoral CD8[+] T cell function. Collectively, all the above results suggest that tumor-derived lanosterol serves as an immune suppressive metabolite that limits the cytotoxic function of CD8[+] T cells by reducing HMGCR protein level and thereby impairing KRAS-ERK signaling.

## DUSP18 is overexpressed in CRC patients and predicts immune deserts

To further study the clinical relevance of the above-described DUSP18-USF1-SREBP2 axis, 20 pairs of human CRC samples (T) and adjacent normal colon tissues (N) were collected and examined for expression of the above proteins and several others. In general, the levels of DUSP18, USF1, SREBP2, HMGCR and LSS were significantly upregulated in CRC samples (Fig. 6a and Supplementary Fig. 8a). All of these except USF1 also positively correlated with DUSP18. Whereas USF1 phosphorylation negatively correlated with DUSP18 and was significantly decreased in CRC samples (Fig. 6b). *DUSP18* mRNA expression was also upregulated in multiple colorectal cancer GEO datasets and some

other cancer types (Fig. 6c). Additional studies revealed that CRCs contained high levels of DUSP18 protein (Supplementary Fig. 8b). Receiver Operating Characteristic Curve analysis indicated that elevated *DUSP18* expression was an effective estimate standard of CRC patient's survival (Fig. 6d). The *DUSP18* gene also tended to be hypomethylated in CRC samples, thus explaining the basis for its overexpression (Supplementary Fig. 8c, d). Finally, high *DUSP18* expression also positively correlated with clinical and TMN stage (Supplementary Fig. 8e).

To further explore the relationship between *DUSP1*8 and the TME of CRC patients, single-cell RNA sequencing data were analyzed from 62 CRC patients (Supplementary Fig. 6f). CD8[+] T cell signatures were significantly increased in tumors with low *DUSP18* expression (Fig. 6e, f and Supplementary Fig. 8g) and *DUSP18* expression negatively correlated with the number of CD8[+] T cells (Supplementary Fig. 8h, i).

Based on classical markers, CD8[+] T cells were reclassified into five subpopulations: exhausted T cells (Tex), effector T cells (Teff), tissue-resident memory T cells, naive T cells, and memory T cells (Fig. 6g). The proportion of Teffs was dramatically upregulated in CRCs with low *DUSP18* expression, while the relative ratio of Texs was lower (Fig. 6h). Gene signatures of CD8[+] T cells also revealed that the IFN-γ and TCR signaling pathways were all enriched in low *DUSP18* expression tumors (Fig. 6i).

Previously reported immune-related gene signatures calculated by the single-sample gene-set enrichment analysis (ssGSEA) algorithm[53], were used to characterize immune-related indicators based on the *DUSP18* expression levels in the TCGA-COAD dataset. Tumor samples with high *DUSP18* expression had lower immune-related indicators scores (Supplementary Fig. 9a–d), lower tumor mutation burden, microsatellite instability, tumor neoantigens and mutant-allele tumor heterogeneity (Supplementary Fig. 9e). In contrast tumor samples with low *DUSP18* expression showed higher enrichment scores of T cell inflammatory gene expression profile, innate anti-PD-1 resistance, immuno-predictive score signatures and PD-1 response signatures (Supplementary Fig. 9f). In addition, *DUSP18* was highly associated with most of the antitumor immunity process (Supplementary Fig. 9g).

To further validate *DUSP18*'s role in regulating tumor immunity in CRC at the transcriptional level, GSEA, GO, and KEGG analyses were applied to TCGA-COAD data. The results showed that low *DUSP1*8-expressing tumor samples were significantly enriched for immune-related functions pertaining to antigen presentation, chemokine signaling, T-cell receptor signaling (Supplementary Fig. 10a-c), whereas high-expressing samples were enriched for cholesterol biosynthetic pathways (Supplementary Fig. 10d). The use of multiple algorithms showed that *DUSP18* expression negatively correlated with infiltrations by CD8[+] T cells and a variety of other immune cells (Supplementary Figs. 10e, f and 11). These findings show that *DUSP18* was significantly upregulated in human CRC samples and was predictive of less immune cell infiltration and the non-inflammatory TMEs.

## Lumacaftor is a potent DUSP18 inhibitor

Given that *DUSP18* plays an important role in tumor immune evasion, we attempted to identify DUSP18 inhibitors to use as potential CRC therapies. Employing a virtual screen based on the AutoDock4 algorithm, the putative DUSP18 inhibitor Lumacaftor was identified as having the highest binding energy to DUSP18 from a library of about two thousand FDA-approved drugs in the ZINC15 compound library (Fig. 7a, b). Lumacaftor has been previously used in the treatment of cystic fibrosis where it normalizes the trafficking of some mutant CFTR proteins to the outer membrane[54]. Based on the crystal structure DUSP18, however, our molecular docking study showed that Lumacaftor interacts with DUSP18 residues Ala105, Ala106, Ser109, Arg110 and Arg142 (Supplementary Fig. 12a). In support of this model, the results of Microscale Thermophoresis

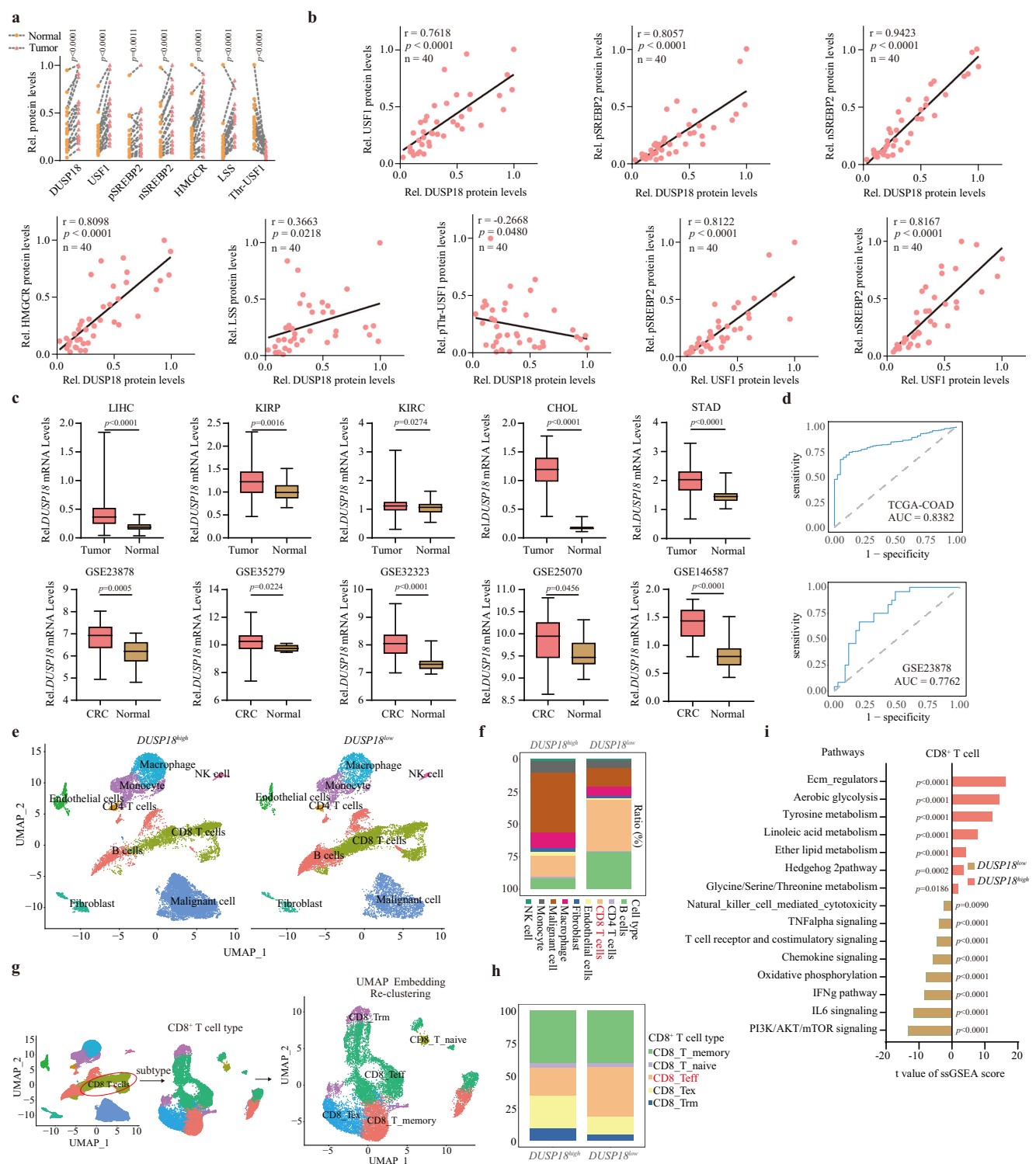

(MST) showed Lumacaftor to have a stronger affinity to WT DUSP18 than to DUSP18 containing mutations in any of the above residues (Fig. 7c). In cellular thermal shift assays (CETSAs), Lumacaftor stabilized the thermal denaturation of the DUSP18 protein in both HCT116 and SW480 cells, thereby indicating that DUSP18 is a direct target of Lumacaftor (Fig. 7d). Lumacaftor also binds to DUSP18 residues that are essential for its phosphatase activity. Unsurprisingly, Lumacaftor treatment reduced the protein and mRNA levels of genes in the cholesterol biosynthesis, while having almost no effect on DUSP18 protein (Fig. 7e, f and Supplementary Fig. 12b, c). In vitro dephosphorylation experiments showed that Lumacaftor inhibited

the ability of purified DUSP18 to dephosphorylate WT USF1 but had no effect on USF1[T100A] (Fig. 7g). In addition, Lumacaftor treatment in sh*DUSP18* CRC cell lines did not further down-regulate the protein levels of USF1 and SREBP2, indicating that the effect of Lumacaftor is DUSP18-dependent (Fig. 7h).

Similar to the effects of *Dusp18* inhibition, Lumacaftor exerted minimal toxicity on cells (Supplementary Fig. 12d–f). Enhanced release of cytokines, elevated cytotoxic capacity and decreased inhibitory molecules were found in CD8[+] T cells co-cultured with the tumor cells with Lumacaftor treatment (Fig. 7i–k). Overall, Lumacaftor was identified as a potent inhibitor of DUSP18 phosphatase activity, of the

**Fig. 6 | DUSP18 is over-expressed in CRC patients and predicts the presence of immune deserts. a** Relative protein level of DUSP18, USF1, pSREBP2, nSREBP2, HMGCR and LSS (tumor samples size: 20, paired normal samples size: 20). Signal intensity of these proteins was quantified by Image J, and then normalized to β-actin band intensity. **b** Correlation between the expression levels of DUSP18 and USF1, pSREBP2, nSREBP2, HMGCR and LSS, as determined by Pearson's $r$ analysis. **c** Differential expression analysis for *DUSP18* in tumor and normal tissues. LIHC, liver hepatocellular carcinoma (tumor samples ($n = 374$), normal samples ($n = 50$)); KIRP, kidney renal papillary cell carcinoma (tumor samples ($n = 289$), normal samples ($n = 32$)); KIRC kidney renal clear cell carcinoma (tumor samples ($n = 535$), normal samples ($n = 72$)); CHOL, cholangiocarcinoma (tumor samples ($n = 36$), normal samples ($n = 9$)); STAD, stomach adenocarcinoma (tumor samples ($n = 375$), normal samples ($n = 32$)). For GSE23878 (tumor samples ($n = 35$), normal samples ($n = 24$)); GSE35279 (tumor samples ($n = 74$), normal samples ($n = 5$)); GSE32323 (tumor samples ($n = 17$), normal samples ($n = 17$)); GSE25070 (tumor samples ($n = 26$), normal samples ($n = 26$)); GSE146587 (tumor samples ($n = 29$), normal samples ($n = 30$)). **d** ROC plot showing the AUC of the *DUSP18* expression from TCGA-COAD ($n = 473$) and GSE23878 ($n = 35$) databases. **e** UMAP plot from high (left, $n = 15$ samples) or low (right, $n = 15$ samples) *DUSP18* expression patients. **f** Bar plot of proportional differences in immune cells between the *DUSP18*^high ($n = 15$ samples) and *DUSP18*^low ($n = 15$ samples) groups. **g** Re-clustering of CD8^+ T lymphocytes, UMAP visualization and marker-based annotation of CD8^+ T lymphocyte subtypes, colored by cluster identity ($n = 12528$ cells). **h** Bar plot of proportional differences in CD8^+ T lymphocytes between the *DUSP18*^high ($n = 2264$ cells) and *DUSP18*^low ($n = 2804$ cells) groups. CD8_T_ memory, memory CD8^+ T cells; CD8_T_ naive, naive CD8^+ T cells; CD8_T_Tex, exhausted CD8^+ T cells; CD8_T_Teff, effector CD8^+ T cells; CD8_T_Trm, tissue-resident memory CD8^+ T cells. **i** Enrichment of different gene signature scores between the *DUSP18*^high ($n = 2264$ cells) and *DUSP18*^low ($n = 2804$ cells) groups in single-cell transcriptomes from re-clustered CD8^+ T cells. Data are presented as mean ± SD (**a**). *P* values were calculated by unpaired two-tailed t-tests (**a**, **c**), modified Fisher's exact tests (**i**). *P* values and R were calculated by Pearson's correlation analysis. Two-sided *P* value was given (**b**); ns, not significant. Source data are provided as a Source Data file.

---

DUSP18-USF1-SREBP2 TF cascade and of cholesterol biosynthesis in CRC cells, with little evidence of toxicity.

## Inhibition of DUSP18 with Lumacaftor sensitizes cancer cells to ICB therapy

Given that Lumacaftor can inhibit DUSP18 activity and enhance the antitumor function of CD8^+ T cells in vitro, the antitumor effect of Lumacaftor were subsequently assessed on tumor models in vivo. The syngeneic mouse MC38 CRC model was utilized to examine how Lumacaftor, with or without the addition of anti-PD-1 antibody, affected tumor growth and survival. Strikingly, combination therapy significantly suppressed tumor growth and prolonged survival of MC38 tumor-bearing immunocompetent C57BL/6 J mice compared to either single-agent or control-treated group (Fig. 8a–c). Analysis of tumor-associated immune cells demonstrated that the combination of Lumacaftor and anti-PD-1 treatment significantly increased the percentage of CD8^+ T cells but had no significant effect on CD4^+ T cells (Fig. 8d, e). Combined treatment also significantly elevated the expression of GzmB and IFN-γ by these CD8^+ T cells (Fig. 8f, g). Similar results were also observed in B16-OVA tumor-bearing immunocompetent C57BL/6J mice with the above-combined treatment (Fig. 8h–j). Flow cytometric analysis confirmed that combination therapy significantly enhanced CD8^+ T cell infiltration in tumor tissue and the expression of GzmB and IFN-γ (Fig. 8k–m). These data were concordant with the results obtained with MC38 tumor studies and strengthened the findings that DUSP18 inhibition with Lumacaftor represses tumor immune evasion and enhances responses to immunotherapy.

## Discussion

CRC has an immunosuppressive TME which prevents the development of an effective response to ICB therapies. There is thus an urgent need to identify the ways reprogram this suppressive TME in order to enhance immunotherapy efficacy. In the current study, we used CRISPR KO screens to discover genes that sensitize CRC to antitumor immunity in host mice that differ in microenvironmental competency. We found that DUSP18 regulates the abundance of the USF1 TF by dephosphorylating it at a specific residue, Thr100. In turn, we have shown that USF1 transcriptionally activates *SREBF2* to mediate lanosterol accumulation in the TME, which suppresses CD8^+ T cell-mediated antitumor immunity. The combination of an anti-PD-1 immunotherapy and Lumacaftor, an FDA-approved small molecule inhibitor of DUSP18, impaired CRC growth in mice and synergistically enhanced antitumor immunity, and better survival in mouse models. We also observed significant associations between *DUSP18* expression, levels of CD8^+ T cell infiltration, and clinical outcomes in human CRCs and published single-cell databases. Our study thus establishes a role of DUSP18 in modulating cytotoxic function of CD8^+ T cell in tumors and in suppressing the efficacy of immunotherapies.

DUSPs are considered to be major regulators of key signaling pathways that are dysregulated in a variety of diseases including cancer. Based on sequence similarity, DUSPs can be categorized into seven subgroups including slingshots, PRLs, Cdc14 phosphatases, PTENs, myotubularin phosphatases, MKPs and atypical DUSPs[55]. Whereas the PTEN and MKP subtypes have been the most intensely studied, little research on other subgroups, particularly the atypical DUSPs, has been performed. By analyzing TCGA-CRC data, we found that DUSP18, one such atypical member, was significantly overexpressed in CRC and was significantly associated with clinical progression in CRC patients.

We found several possible and non-mutually exclusive explanations for DUSP18's upregulation in CRCs. These included epigenetic alterations, especially those involving *DUSP18* gene promoter hypomethylation, which correlated with its expression levels (Supplementary Fig. 8c, d). A second explanation involved the direct upregulation of *DUSP18* by oncogenic TFs. Some of these, including c-Myc or STAT3, may directly bind to the *DUSP18* promoter and upregulate expression. Finally, somatic mutations in or amplification of the *DUSP18* gene may also lead to its upregulation in tumors.

Single-cell RNA-Seq data analyses also found that high *DUSP18* expression by tumors significantly negatively correlated with CD8^+ T cell infiltration and activation and positively correlated with TME-associated CD8^+ T cell exhaustion. Further analysis of bulk RNA-Seq data in TCGA-COAD, found that *DUSP18* expression negatively correlated with immune infiltration-associated signal sets and positively correlated with immunosuppressive signals. Using combined analysis of multiple algorithms, we determined that high expression of *DUSP18* in tumors predicted prognostically worse CD8^+ T cell infiltration scores. Tumors with high *DUSP18* expression are more inclined to be representative of "immune deserts", which are generally thought to be immunotherapy-resistant. These findings, based on clinical and genomic data, may provide useful biomarkers for personalizing treatment strategies.

Numerous studies have elucidated the pivotal roles of tumor metabolic reprogramming in driving tumor proliferation and facilitating immune evasion[56,57]. The re-wiring of cholesterol metabolism documented herein serves as one such example whereby this pathway is co-opted to generate an immunosuppressive TME. It may include the delivery of tumor-derived cholesterol to myeloid-derived stem cells (MDSCs) through small extracellular vesicles[36]. However, roles for cholesterol metabolism intermediates have largely focused on their effects on tumor proliferation. For example, mevalonate kinase can promote tumor growth by stabilizing mutant p53 proteins[58].

Our study found that, in cancer cells, DUSP18 affects cancer progression through its influence on USF1-SREBP2-driven

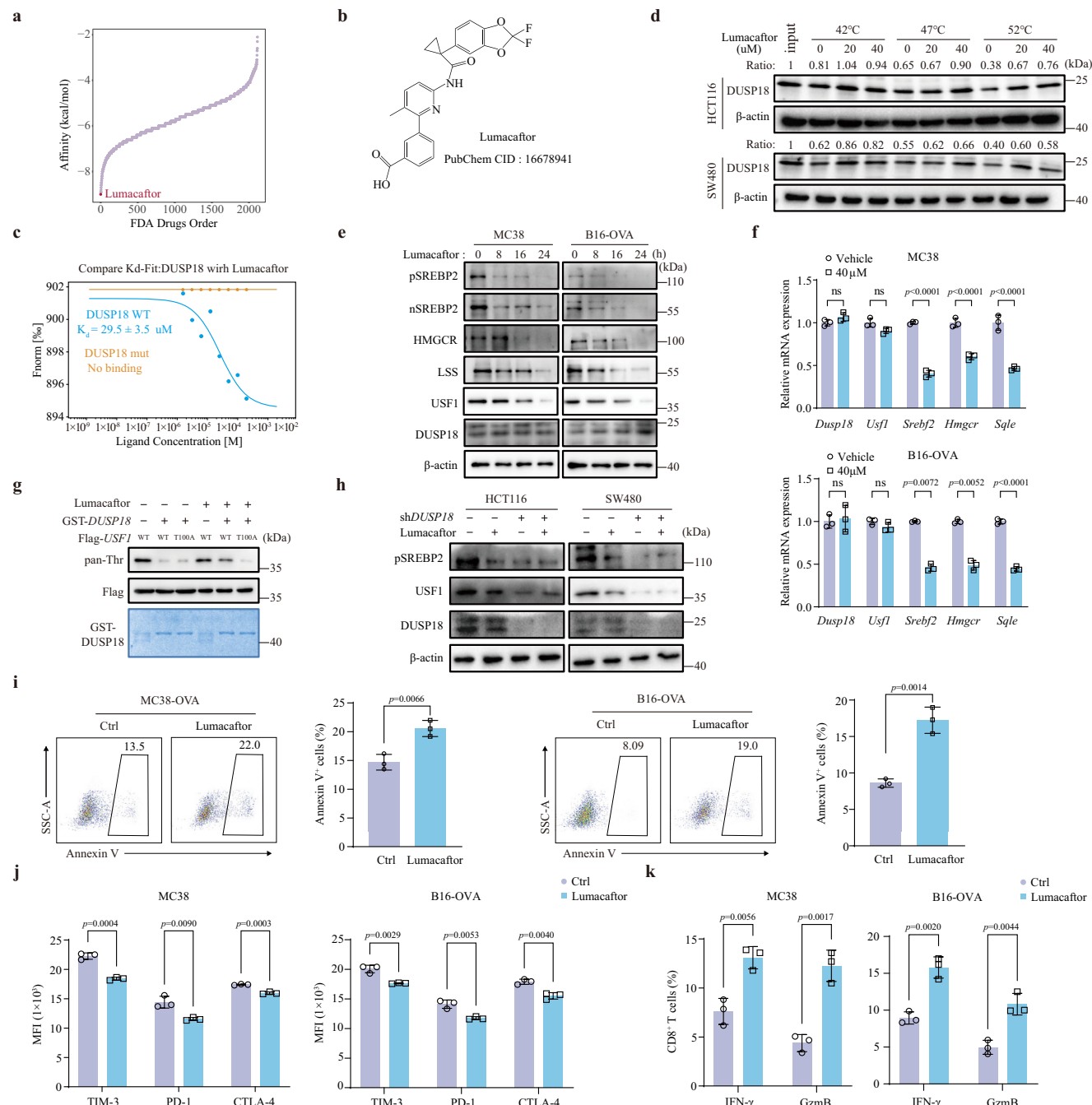

**Fig. 7 | Lumacaftor is a potent DUSP18 inhibitor. a** Affinity rank of small molecules with DUSP18 according to virtual screening. **b** Chemical structure formula of Lumacaftor. **c** The equilibrium dissociation constant ($K_d$) value was determined as the binding of Lumacaftor to purified human DUSP18 or DUSP18 mutant (A105/106D, S109D, R110D and R142D) proteins using MST ($n = 3$). **d** Effect of Lumacaftor on the thermal denaturation of cellular DUSP18 protein. CETSA was performed on cell lysates from HCT116 and SW480 cell lines. **e** Western blot for the indicated proteins in MC38 and B16-OVA cell lines following exposure to 40 µM Lumacaftor for 24 h. **f** mRNA levels of indicated genes from MC38 and B16-OVA cells treated by control or 40 µM Lumacaftor for 24 h were analyzed using RT-qPCR ($n = 3$). **g** Recombinant GST-*DUSP18*-His and Flag-*USF1* (WT or T100A) were used for in vitro dephosphorylation assay, with or without the addition of 40 µM Lumacaftor. The levels of USF1 phosphorylation were measured by immuno-blotting analysis. **h** Lumacaftor was added to shCtrl or sh*DUSP18* HCT116 and SW480 cells, and the related protein levels were detected by immunoblot. **i** Quantification of apoptosis in MC38-OVA and B16-OVA cells that were pretreated with 40 µM Lumacaftor for 24 h and co-cultured with CD8+ T cells for 24 h ($n = 3$). **j** Mean Fluorescence Intensity (MFI) quantification of PD-1, TIM-3, CTLA-4 in CD8+ T cells co-cultured with MC38 and B16-OVA cells that were pretreated with 40 µM Lumacaftor for 24 h ($n = 3$). **k** Quantification of cytokine production by CD8+ T cells co-cultured with MC38 and B16-OVA cells pretreated with 40 µM Lumacaftor for 24 h ($n = 3$). Data are presented as mean ± SD (**f**, **i**–**k**). *P* values were calculated by unpaired two-tailed t-tests (**f**, **i**–**k**); ns not significant. All IB data are representative of three independent experiments. Source data are provided as a Source Data file.

transcriptional cascade that increases micro-environmental lanosterol and blocks CD8+ T cell activation. Lanosterol accumulation in the TME hampers T cell's HMGCR protein level, consequently diminishing the availability of isoprenoids, which are essential for the post-translational prenylation modifications of KRAS and for CD8+ T cell activation[59].

Notably, tumor cells defective in *DUSP18* or lanosterol do not present growth limitations in vitro or in immunodeficient mice,

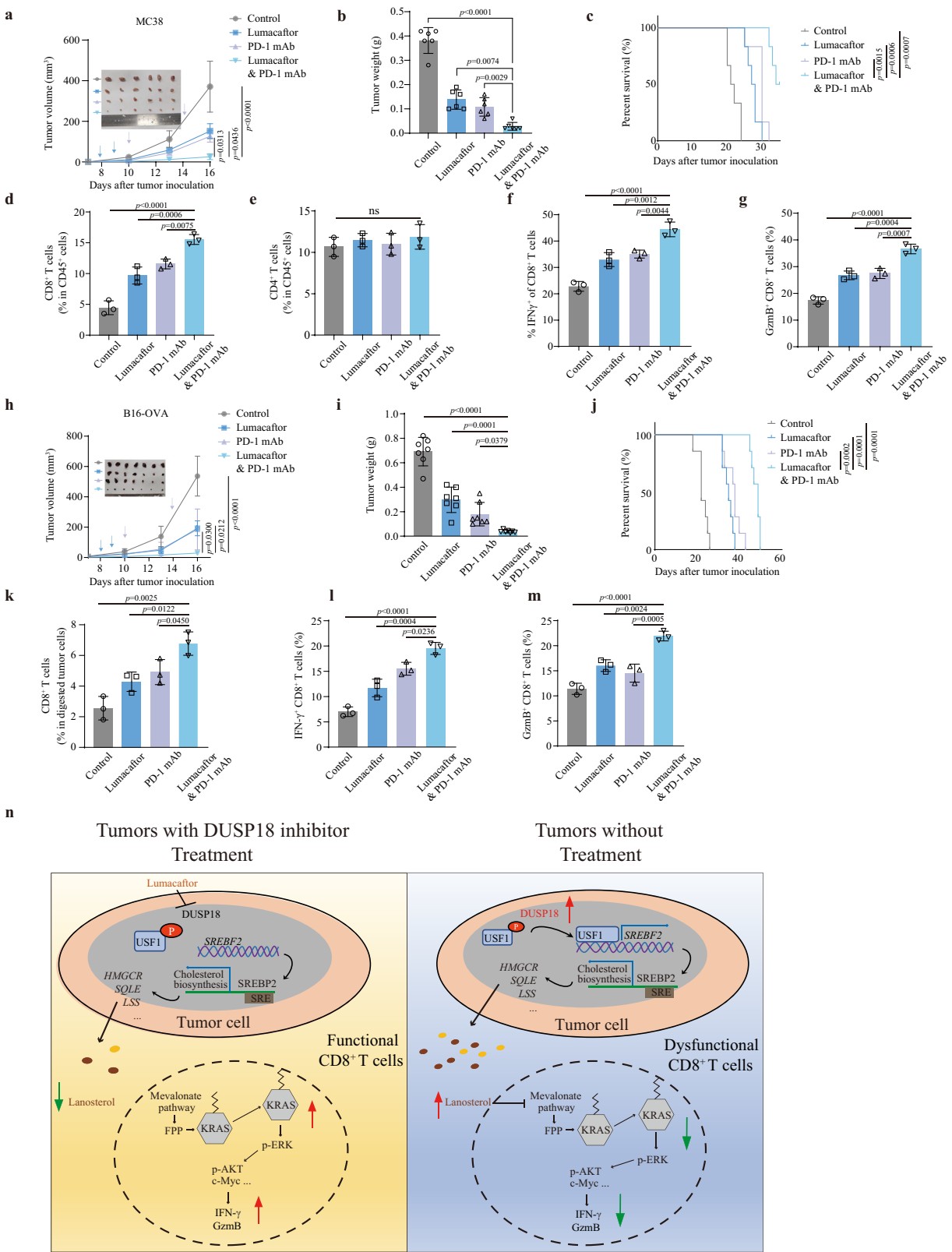

**Fig. 8 | Targeted inhibition of DUSP18 with Lumacaftor sensitizes cancer cells to ICB therapy.** Therapeutic effect of Lumacaftor on tumorigenesis of MC38 cells or B16-OVA cells in C57BL/6J mice. 4–6 weeks C57BL/6J mice were injected i.p. with 30 mg/kg Lumacaftor on days 7, 10, and 13, and 100 μg anti-PD-1 on days 7 and 10 following subcutaneous injection of tumor cells. **a** (*n* = 6), **h** (*n* = 7), Tumor growth curves. **b** (*n* = 6), **i** (*n* = 7), Tumor weights. **c** (*n* = 6), **j** (*n* = 7), Survival curves. **d** (*n* = 3), **k** (*n* = 3), Percentage of tumor-infiltrating CD8⁺ T cells determined by flow cytometry. **e** Percentage of tumor-infiltrating CD4⁺ T cells (*n* = 3). **f** (*n* = 3), **l** (*n* = 3), Percentage of IFN-γ⁺ CD8⁺ T cells in the TME. **g** (*n* = 3), **m** (*n* = 3), Percentage of GzmB⁺ CD8⁺ T cells in the TME. **n** Schematic model of the regulatory pathway and mechanism of DUSP18 in tumor immune evasion. Data are presented as mean ± SD (**a**–**m**). *P* values were calculated by unpaired two-tailed *t* tests (**a, b, d, f, g, h, i, k**–**m**), one-way ANOVA (**e**) or log-rank (Mantel-Cox) test (**c, j**). ns not significant. Source data are provided as a Source Data file.

indicating that *DUSP18* or lanosterol is not necessarily required for tumor cell growth and thereby further supporting the idea that its function in the context of cancer is an immunomodulatory one. However, it has been reported that inhibition of lanosterol production could inhibit the proliferation of hepatocellular carcinoma[60], glioma[61], and pancreatic cancer tumors[62]. It also has been reported that dietary lanosterol significantly suppresses the formation of aberrant colonic crypts[63] and inhibits hormone-dependent growth of breast cancer cells[64,65].

Due to the great heterogeneity of different tumors, the mechanism of action of lanosterol may well differ in different contexts. The essentiality of lanosterol for tumor growth depends on the dependence of tumor cells on lanosterol and the amount of lanosterol in the cells. For tumors that are heavily dependent on cholesterol synthesis, its inhibition and that of lanosterol as well, is certainly likely to inhibit their growth. Increasing proliferative rates by the overexpression of oncogenes such as *Myc* or the loss of tumor suppressors such as *TP53* my increase the demands on the biosynthetic pathway that provides both cholesterol and lanosterol. In cases where tumor growth is less robust, the inhibition of lanosterol may not significantly affect cell viability per se, possibly because the level of lanosterol is too low to significantly impact the TME. At least in CRC, asynchronous alternation of enzymes along the pathway is commonly observed. One of the rate-limiting enzymes for cholesterol synthesis, HMGCR, as well as others including DHCR24 and IDI1 did not differ between tumor and normal tissues (Supplementary Fig. 12g), which makes it appear that CRC is not overly dependent on the cholesterol synthesis pathway. However, SQLE and LSS, as key enzymes for the synthesis of lanosterol and distal cholesterol products were significantly upregulated (Supplementary Fig. 12h). This provides an excellent example of the asynchronous upregulation mentioned above and involving the distal portion of the cholesterol biosynthetic pathway that includes lanosterol. This suggests that lanosterol over-production by some CRCs is not meant to specifically satisfy any growth-related needs but is rather intended to support immune evasion. These results suggest that the dependence on lanosterol in some cases of CRC, is relatively low as far as it relates to the growth of the tumor cells. This could explain why *DUSP18* inhibition-mediated reduction of lanosterol and its downstream products does not affect the proliferation of the tumors. In such a scenario, the importance of lanosterol would be highlighted only in an immunocompetent system. Considering this, we suggest that future pre-clinical studies aimed at targeting DUSP18 in cancer therapy be conducted in immune-competent backgrounds.

Taken together, our observations have revealed a crucial role for DUSP18/USF1/SREBF2-lanosterol signaling in tumor immunosuppressive reprogramming. Pharmacologically targeting this signaling pathway could reinforce antitumor immunity, particularly in tumors where the latter alone initially appears to be of minimal value.

## Methods

Our research complies with all relevant ethical regulations of Wuhan University. Animal studies were approved by the Institutional Animal Care and Use Committee of Wuhan University and complied with all relevant ethical regulations (WAEF-2022-0060). The mice with orthotopic tumors, authorized by the Committees on Animal Research and Ethics, consistently follow the humane endpoint. If the animal starts showing signs of immobility, a huddled posture, the inability to eat, ruffled fur, or self-mutilation, the animal will be euthanized immediately. The maximal tumor volume permitted by the Institutional Animal Care and Use Committee of Wuhan University is 2000 mm³ (WAEF-2022-0060). Thus, when tumor volumes reached a maximum of 2000 mm³, the mice were immediately euthanized. In mouse subcutaneous graft tumor experiments, both male and female mice were used and were randomly distributed and assigned to each group.

CRC tissues from patients were obtained from the Department of Colorectal and Anal Surgery at Zhongnan Hospital of Wuhan University, Wuhan, China. The use of pathological specimens and the review of all pertinent patient records were approved by the Ethics Committee of Wuhan University (2022030). Informed consent was obtained by participants.

### Cell culture

B16-F10 was kindly provided by Prof. Jinfang Zhang (Wuhan University). CT26 was generously gifted by Prof. Junjie Zhang (Wuhan University). All other cell lines were originally purchased from American Type Culture Collection (ATCC). All cell lines were tested for mycoplasma contamination and no cell lines were contaminated. All cells were cultured in Dulbecco's modified Eagle's medium (DMEM; Gibco) supplemented with 10% fetal bovine serum (FBS, Gibco), 1% Penicillin–Streptomycin Solution (Beyotime) at 37 °C in 5% $CO_2$.

### Plasmid construction and establishment of stable cell lines

shRNAs against DUSP18 and USF1 were constructed into the pLKO.1 vector (Sigma-Aldrich). The target sequences of all shRNAs used in this study are summarized in Supplementary Table 1. DUSP18, USF1 and their corresponding mutations were subcloned into the pHAGE-CMV-MCS-PGK-3×Flag, pCMV-HA, or pLKO.1-GFP empty vector, the PCR primer sequence used in this study is summarized in Supplementary Table 2. The lentivirus vectors were co-transfected with psPAX2 plasmid and pCMV-VSV-G plasmid into HEK293T cells using PEI. Culture medium containing virus particles was collected 48 h post-transfection and added into the culture medium of tumor cells with 8 μg/ml polybrene following the selection with 2 μg/ml puromycin.

### Animals

All animal procedures were approved by the Animal Care Committee of Wuhan University. *Dusp18^flox/flox* mice and *Villin^Cre* were purchased from Cyagen (Guangzhou, China). All mice were housed in a specific-pathogen-free animal facility at Wuhan University and were maintained in a 12 h light/12 h dark cycle, and the housing temperature and humidity were maintained at 24 °C and 50%, respectively. For AOM/DSS-induced colorectal cancer in mice, 8-week-old C57BL/6J WT mice and conditional knockout (CKO) mice were injected intraperitoneally with 10 mg/kg AOM (Sigma-Aldrich). 7 days later, mice were given drinking water containing 2.5% DSS (MP Biomedicals, Santa Ana, CA, USA) for 7 days followed by 2 weeks of regular drinking water for recovery. This same cycle was repeated twice. On day 110, tumor burdens were evaluated. All mouse genotype identification primer sequences used are shown in Supplementary Table 5.

### Viral library production

Mouse CRISPR Deletion Library generated against ~2000 genes encoding drug targets, kinases and phosphatases and containing ~22,000 gRNAs[39] was from Addgene (Catalog# 1000000122). CRISPR library plasmids were transfected into HEK293T cells at 80% confluence in 15 cm tissue culture plates. Viral supernatants were collected at 48 h and 72 h post-transfection and passed through a 0.45 μm pore size hydrophilic PVDF membrane. The supernatant was then aliquoted and stored at −80 °C until use.

### In vivo CRISPR screens in MC38 tumor cells

We first constructed an MC38 cell line stably expressing Cas9. To perform the CRISPR screens, we transduced ~2 × 10⁷ MC38-Cas9 cancer cells with lentivirus containing the above library at a multiplicity of infection of ~0.3. After 7 days of puromycin selection, cells were assigned for ex vivo experiments. Transduced mouse cancer cells were injected subcutaneously into nude mice and C57BL/6J wild-type mice. MC38 cells transduced with libraries were also grown in vitro at ~1000× library coverage for the same time period as the animal experiment.

Mice were euthanized 14–16 days after tumor implantation, and PCR was performed on genomic DNA from harvested tumors to amplify the sgRNA regions for next-generation sequencing. Sequencing data were analyzed by using MAGeCK (v.0.5.6) and MAGeCK-VISPR (v.0.5.3) to identify sgRNAs that were significantly enriched or depleted in any conditional comparison[66].

## Cell viability assays

Tumor cells were seeded in 96-well plates which were cultured for 4 days prior to cell counting which was performed on at least three biological replicates using a Cell Counting Kit 8 following the manufacturer's protocol.

## Co-IP and immunoblot

Cells were lysed with 1 ml RIPA buffer [50 mM Tris-HCl (pH 7.4), 150 mM NaCl, 5 mM EDTA, 1% Nonidet P-40, 0.5% sodium deoxycholate, 10 µg/ml aprotinin, 10 µg/ml leupeptin, and 1 mM phenylmethylsulfonyl fluoride]. 950 µL cell lysate was mixed with the indicated antibody and beads and incubated overnight at 4 °C. The beads were collected in a magnetic holder and washed three times with 1 ml of RIPA lysis buffer. Immunocomplexes were disrupted with 4×SDS Loading buffer, heated to 95 °C for 10 min and examined by immunoblotting using the indicated antibodies[67].

## Tumor growth and treatment

$1 \times 10^6$ shCtrl or sh*Dusp18* MC38 or B16-OVA tumor cells were trypsinized, washed, re-suspended in DMEM and subcutaneously injected into the flanks of mice. Tumor volumes were measured by length (a) and width (b) and calculated as Tumor Volume = $a \times b^2/2$. For anti-CD8α and anti-PD1 treatments, 100 µg of antibody was diluted in PBS and injected into tumor-bearing mice intraperitoneally on the indicated days. For Lumacaftor treatment, DMSO-resolved drug was diluted in corn oil and injected into tumor-bearing mice intraperitoneally with a dosage of 30 mg/kg on the indicated days.

## Flow cytometry

shCtrl or sh*Dusp18* MC38 or B16-OVA tumor tissues were minced with scissors and digested at 37 °C 1 h in DMEM containing 5% FBS, 1 mg/ml Collagenase IV and 0.1 mg/ml DNase I. Single cell suspensions were filtrated through 70 µm cell strainers and washed twice with Staining buffer (PBS containing 2% FBS and 1 mM EDTA). Cells were re-suspended in the staining buffer and stained with following antibodies on ice for 30 min: APC/CY7 anti-CD45 Antibody (BD Biosciences; Cat# 157618), FITC anti-mouse CD8a antibody (Biolegend; Cat# 100705;), APC anti-mouse CD4 antibody (Biolegend; Cat# 100515), PE anti-PD-1 antibody (eBioscience; Cat# 12-9985-82), PE/Cyanine7 anti-mouse CTLA-4 antidoy (BD Biosciences; Cat# 106313), BV650 Mouse Anti-Mouse CD366 (TIM-3) (BD Biosciences; Cat# 747623), APC-MHC Class I (H-2Kb) antibody (eBiosciences; Cat# 17-5958-80), APC- OVA257-264 (SIINFEKL) peptide bound to H-2Kb antibody, (eBiosciences; Cat# 17-5743-82), APC anti-mouse Granzyme B (eBiosciences; Cat# 17-7311-82) and PE anti-mouse IFN-gamma (eBiosciences; Cat# 12-8898-82). Gating and sorting strategies were provided in Supplementary Fig. 13. The reagents and antibodies used in this study are listed in Supplementary Table 3 and Supplementary Table 4. For assessment of apoptosis, cells were treated with Annexin V-FITC/PI Apoptosis Detection Kit (Yeasen) according to the manufacturer's instructions. The apoptotic cells were then analyzed via FlowJo software (v10.8.1).

## Intracellular cytokine staining

For intracellular staining, tumor-infiltrating lymphocytes (TILs) were purified from single cell suspensions of tumors using 40% Percoll gradients. $1 \times 10^6$ TILs were stimulated with Phorbol 12-Myristate 13-Acetate (PMA) (50 ng/ml) and ionomycin (500 ng/ml), and blocked with Brefeldin A (1:1000) for 4 h at 37 °C. After a washing step, cells were stained with anti-CD45 and anti-CD8 for 30 min on ice, fixed and permeabilized with eBioscience™ Intracellular and Permeabilization Buffer (eBioscience) on ice for 15 min, and then washed twice with Staining buffer. Anti-IFN-γ and anti-Granzyme B antibodies were added and incubated for 1 h on ice and analyzed by flow cytometry.

## T cell and tumor cell co-culture assay

Mouse naive CD8+ T cells were isolated from spleen of C57BL/6 mice using the EasySep mouse CD8+ T cell isolation kit (STEMCELL) according to the manufacturer's protocol and were then immediately activated with anti-CD3/CD28 antibody (Biolegend) in RPMI-1640 containing 10 ng/mL mouse IL-2 (MCE), 10% FBS (Gibco), 1% Penicillin-Streptomycin Solution (Beyotime) and 40 µM β-Mercaptoethanol (Gibco). T cells were stimulated in vitro for at least 3 days before being co-cultured with tumor cells. $1 \times 10^5$ MC38 or B16-OVA tumor cells were seeded into 24-well plates with DMEM complete medium. In vitro activated CD8+ T cells were then added and co-cultured with tumor cells at a ratio of 1:1 for 24 h. 4 h before cell collection, Brefeldin A (eBioscience, 1:1000) was added to block cytokine secretion. T cells were washed and re-suspended in staining buffer and stained with anti-CD8 antibodies (eBioscience) for 30 min on ice. After an additional washing step, intracellular staining was performed as previously described using the intracellular cytokine staining protocol. For CD8+ T cell killing assays, shCtrl or sh*Dusp18* MC38-OVA and B16-OVA cells were co-cultured with activated OT-I T cells at a ratio of 1:1 for 48 h. Apoptotic cells were quantified using the Annexin V-FITC/PI Apoptosis Detection Kit (Yeasen) according to the manufacturer's instructions and analyzed by Beckman Cytoflex. LDH release was determined using LDH Cytotoxicity Assay Kit (Yeasen) following the manufacturer's instructions.

## RNA-seq and bioinformatic analysis

Total RNAs were isolated from $1 \times 10^7$ shCtrl or sh*Dusp18* MC38 and HCT116 cells by using the TRIzol reagent following the manufacturer's instructions (TransGen Biotech). RNA libraries were constructed by the BENAGEN company (Wuhan) and sequenced on the Illumina NovaSeq platform. RNA-seq raw data was detected and controlled by using FASTQC software (v 0.11.9) and aligned to the genome GRCm39 or GRCh38 by HISAT2 software (v2.2.1) and FeatureCounts (v1.28.1) for calculating gene counts. Differentially expressed genes (DEGs) were identified with $p$ value < 0.05 and absolute $log_2$ fold-change >0.585 by DESeq2 R package (v1.16.1) and plotted with R packages ggplot2 (v4.2.3). Heatmaps were generated using GraphPad Prism 8.0 version or Pheatmap package (v1.0.12). Gene Ontology (GO) and Kyoto Encyclopedia of Genes and Genomes (KEGG) enrichment analyses were performed by the R packages clusterProfiler (v4.0.5). Gene-set enrichment analyses (GSEA) were performed using the R packages GSEABase software (v1.54.0).

## Cholesterol metabolomics analysis of CRC TMEs

The LC-MS sample preparation protocol, including sterol extraction, hydrolysis, derivatization, and sample cleaning[68]. For mice CRC interstitial fluid, 200 µL of mice CRC interstitial fluid was taken which was isolated from all tumor samples at $106\,g$[69], and 800 µL of extraction solvent (DCM: MeOH = 2:1, v/v) containing 6.5 mg BHT was added for sterol extraction. The prepared samples were analyzed by using an Agilent DTIM-QTOFMS 6560 coupled with an Agilent UHPLC 1290 (Agilent Technologies).

## Reverse transcription-quantitative PCR (RT-qPCR) analysis

Total RNAs were extracted using the TRIzol reagent (TransGen Biotech), and reverse transcription reactions were performed using the MonScript™ RTIII All-in-One Mix with dsDNase reagent kit (Monad biotech) and oligo(dT) primers. The RT-qPCR system was used 2X Universal SYBR Green Fast qPCR Mix (ABclonal) and was performed

with the Bio-Rad CFX Detection System (Bio-Rad). The housekeeping gene, *ACTB*, was used as a normalization control. All primers used are listed in Supplementary Table 6.

### Small molecule inhibitor screening and molecular docking
The human DUSP18 crystal structure (PDB:2ESB) was selected as a receptor. FDA-approved drugs from the ZINC15 data base (https://zinc15.docking.org) were chosen as ligands, Autodock vina software (v.1.1.2) was used to perform the docking procedure[70]. Docking results were visualized with pymol software (v2.4.0 Open-Source).

### Microscale thermophoresis
A Monolith NT.115 microscale thermophoresis instrument (Nano-Temper Technologies, Germany) was used to measure the Kd value of binding of Lumacaftor to DUSP18. All cell samples to be assayed were lysed with RIPA buffer. Protein samples and different concentrations of drugs were mixed, incubated at room temperature for 5 min and then added to silica capillaries for machine detection. Data were analyzed with Nano-Temper Analysis software (v.2.3).

### Cellular thermal shift assay
These studies were conducted according to a previously described protocol[71,72]. Briefly, $5 \times 10^6$ cells were pretreated with or without 40 μM Lumacaftor for 24 h before being used for cellular thermal shift assays. After collection, the cells were chilled on ice, washed with cold PBS containing protease inhibitors and transferred to PCR tubes in 100 μL PBS. The cells were then heat-shocked in a Bio-Rad T100 thermal cycler at the indicated temperature (40–60 °C) for 3 min to denature the proteins, returned to room temperature for 3 min and then cooled on ice. The cells were lysed by three freeze-thaw cycles with liquid nitrogen and 25 °C water bath and centrifuged at 12,000 rpm, 4 °C for 10 min. The supernatant was then boiled with 4×SDS loading buffer for western blotting. The bands were quantified using the Image J software.

### Analysis of single-cell RNA-seq and TCGA data
Single-cell RNA-seq data were obtained from public dataset (GSE178341) in Gene Expression Omnibus (GEO) and analyzed using the Seurat package in R[73]. Each sample was individually quality checked. Cell screening criteria were as follows: at least 300 detected genes with no more than 10% mitochondrial reads. Genes expressed in fewer than five cells for individual samples were filtered. Multiple single-cell sample integration and batch effect correction were performed using the harmony algorithm. Gene characterization of 186 metabolic and signaling pathways collected from the MSigC2 database was pooled. Single-cell characterization scores were obtained using genomic variation analysis (GSVA) and the GSVA software package from Bioconductor. The differential metabolic and signaling pathways between the *DUSP18*-high and *DUSP18*-low groups were calculated using the limma package.

TCGA Transcriptome data and clinical information of solid tumors were downloaded on the UCSC platform (https://xenabrowser.net/datapages/). ENSEMBL ID was mapped to gene symbol with R package clusterProfiler (v4.0.5). Gene expression was further normalized to TPM (transcripts per kilobase million) values based on sequencing depth and the longest transcript length. The cytotoxic T lymphocyte (CTL) score was defined as the average expression of five reported signature genes (*CD8A*, *CD8B*, *GZMA*, *GZMB* and *PRF1*) to reflect the cytotoxicity of tumor-infiltrating T cells. R package GSVA was used to analyze the composition of tumor-infiltrating immune cells (including NK cell, activated CD8$^+$ T cell, activated CD4$^+$ T cell et al). Correlation between *DUSP18* mRNA levels and CTL scores were calculated by Pearson's algorithm. The list of publicly available gene signatures selected in this study is shown in Supplementary Data 6.

### Statistics and reproducibility
Statistical analysis was performed using GraphPad Prism 8.0. For comparison of two groups, two-tailed unpaired Student's *t* tests or Mann-Whitney *U* tests were performed. For comparison of more than two groups, one-way ANOVA was performed. For Kaplan-Meier survival curves, the *p* values were calculated using the log-rank test. Data are shown as mean ± SD. The correlation was analyzed using a Pearson correlation test. $P < 0.05$ were considered significant, and statistical significance was denoted with exact *p* value and ns, not significant ($p > 0.05$). Each experiment was repeated independently at least three times and with similar results.

### Reporting summary
Further information on research design is available in the Nature Portfolio Reporting Summary linked to this article.

## Data availability
The RNA-seq data generated in this study have been deposited in the Gene Expression Omnibus (GEO) database under the accession number GSE264145. The mass spectrometry proteomics data of MC38 cells generated in this study have been deposited to the ProteomeXchange Consortium database (http://proteomecentral.proteomexchange.org) under accession code PXD053284. The CRISPR screens data are provided in Supplementary Data 1. The cholesterol metabolomics data are provided in Supplementary Data 5. The transcriptomic data and methylation data used in this study are available in the CRC cases in The Cancer Genome Atlas (TCGA) database (https://portal.gdc.cancer.gov/). A public single-cell RNA-seq data were available in the GEO database under the accession number GSE178341. The transcription factor binding site prediction was performed online with the JASPAR database (https://jaspar.genereg.net/) and humanTFDB database (http://bioinfo.life.hust.edu.cn/HumanTFDB). A public USF1 ChIP-seq data were available in the GEO database under accession number GSE32465. The remaining data are available within the Article, Supplementary Information or Source data file. Source data are provided with this paper.

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

## Acknowledgements

The authors thank Profs. Jinfang Zhang (Wuhan University, Wuhan, China) for B16-F10 cells and Congqing Jiang (Wuhan University, Wuhan, China) for CRC patient samples. This work was supported by grants from the National Nature Science Foundation of China (32270828, 92057108, 81772609 to Y.L.); the Special Foundation for Major Science and Technology Program of Hubei Province (2022ACA005 to Y.L.); the Fundamental Research Funds for the Central Universities (2042022dx0003, 2042021kf0229 to Y.L.); Sino-foreign Joint Scientific Research Platform Seed Fund of Wuhan University (WHUZZJJ202204 to Y.L.); Technical support from Instrument Platform Center (Medical Research Institute, Wuhan University, Wuhan, China) and support from Mass Spectrometry Platform (College of Life Science, Wuhan University, Wuhan, China).

## Author contributions

X.Z. and Y.L. designed the study; X.Z. performed most of the experiments; X.Z., G.W., C.T., and L.D. constructed plasmids; X.Z. performed to analyze sequencing and clinical data; X.Z., G.W. performed animal experiments. All authors discussed the results. X.Z., E.P., and Y.L. wrote the manuscript with comments from all authors.

## Competing interests
The authors declare no competing interests.
