## [Peer Review File · Nature Communications]

Inhibition of DUSP18 impairs cholesterol biosynthesis and promotes anti-tumor immunity in colorectal cancerREVIEWER COMMENTS

Reviewer #1 (Remarks to the Author): with expertise in colorectal cancer

In this manuscript, Zhou et al. reported that the DUSP18-USF1-SREBP2-driven transcriptional cascade is involved in CRC immune escape by increasing micro-environmental lanosterol. The CRISPR screening in vivo implicates that depleted of DUSP18 reduced tumor growth rates, which correlate with high levels of CD8+ T cell activation. Mechanistically, DUSP18 dephosphorylates and stabilizes USF1, which induces Srebp2 to accumulate lanosterol and release it into TME. Lanosterol absorbed by CD8+ T cells and reduces RAS protein prenylation and function to inhibits their activation. Finally, the combination of anti-PD-1 antibody and Lumacaftor inhibited CRC growth in mice and synergistically enhanced anti-tumor immunity.

This is a well-performed study. Basically, the data is extensive, detailed and clear. However, the following issues should be addressed:

1. In the introduction, the authors extensively discussed CTL immune infiltration and cholesterol metabolism, and then mentioned the relationship between phosphatase and immune infiltration. According to the experimental results, the target gene DUSP18 was first screened from the phosphatase CRISPR library, and then the enrichment analysis focused on cholesterol metabolism. Therefore, it is suggested to change the order of presentation.
2. The strategy of CRISPR screening in Fig.1a was performed in mice, including Nude mice and C57BL/6J mice. However, it is better to perform genomic and phenotypic analysis of the two strains to identify differences that may influence their rates of tumor growth.
3. In the preliminary screening in Fig.1, it was found that the expression of DUSP18 in CRC tissues was negatively correlated with CTL immune infiltration, indicating that DUSP18 may related with immune suppression in CRC. However, in the animal experiments of Nude mice and C57BL/6J mice, only Dusp18-knockdown cells and WT cells were compared, and the experiment should be supplemented that cells with Dusp18 overexpression were inoculated into mice to observe the CTL infiltration and tumor growth.
4. In the results of Fig.2, RNA-seq was performed using control and DUSP18-knockdown HCT116 cells to establish the molecular mechanisms of DUSP18 in suppressing anti-tumor immunity. However, it is better to perform the transcriptional sequencing in tumor cells of

mice such as MC38 due to the underlying genetic mechanisms of immune suppression may difference in these two strains of cell lines.

5. In line 210, they hypothesized that DUSP18 regulates a TF that is involved in cholesterol synthesis pathway. However, they found that SREBP2 was responded to DUSP18 knockdown, without any screen strategy. How about other TFs involved in cholesterol synthesis pathway? Do they also respond to DUSP18 knockdown?

6. In Fig.3, they identified that USF1 knockdown declined the expression of SREBP2. It is better to perform the USF1overexpression experiments.

7. In Fig. 4, only the Dusp18-Knockdown MC38-OVA model was established, but no Dusp18-overexpression or rescue experiments.

8. In Fig. 5e, the authors cultured primary CD8+ T cells with different concentrations of exogenous lanosterol. It is better to add an inhibitor to block the endogenous lanosterol as a reverse experiment.

9. It is suggested that the gene names should be consistent in the same sentence. In the text, when they talking about the same species, the same sentence often mixes human gene like DUSP18 and mouse gene like Dusp18.

10. It is suggested that the Figures in the text should be arranged according to the order of mention.

11. The language in the manuscript is acceptable, but further language modification is needed.

Reviewer #2 (Remarks to the Author): with expertise in cancer immunology, CRISPR screens

In the manuscript by Zhou X et al. entitled “Colo-Rectal Cancer Immune Escape is Mediated by a DUSP18-USF1-SREBP2-Driven Transcriptional Cascade that Increases Micro-environmental Lanosterol and Blocks CD8+ T Cell Activation”, the authors utilized a genetical screen approach to identify the immunological role of tumor intrinsic DUSP18. Silencing Dusp18 in murine colon tumors improves tumor infiltration and effector function of CD8+ T cells. Mechanistically, DUSP18 dephosphorylates Thr100 of USF1, then stabilizes USF1 for transcriptionally activating SREBP2, a gene involved in cholesterol biosynthesis. Ultimately, upregulated DUSP18 in tumors result in accumulation of lanosterol in tumor

microenvironment. Increased lanosterol can impair CD8+ T cell tumor infiltration and function. These results reveal a new tumor immune evasion mechanism via the DUSP18-USF1-SREBP2-lanosterol axis. Finally, the authors also identified Lumacaftor as a potent DUSP18 inhibitor by in silico drug screen and demonstrated the potential of Lumacaftor sensitizes colon tumors to immunotherapy. The significance and novelty of this paper are high. The manuscript includes sufficient evidence to support findings based on their own working models and appropriate references to credit previous work. The quality of this manuscript can be further improved from the following aspects. Specifically:

1. For the CRISPR screen, the authors have chosen the MAGeCK algorithm to analyze their screening results. The assumption of RRA algorithm from MAGeCK is that if a gene has no effect on selection, then sgRNAs targeting that gene should be uniformly distributed across the ranked list of all the sgRNAs. However, the sub-CRISPR library occupies biased distribution during the screening process because of the target gene selection. Therefore, another algorithm should be employed. Also, the authors should provide more information about the genes included in the sub-CRISPR library for better understanding.
2. The authors used two lines from different species to explore the role of DUSP18. The identification of DUSP18 was used a murine line, but the mechanistic studies in Fig 3 were used the human line, HCT116. At least, the authors should consider confirming the identified transcriptional changes in MC38 tumors.
3. Their preclinical findings related to antitumor immune responses are largely dependent on OVA-expressing tumor cells. As OVA is a foreign antigen, these models might not well represent immune responses against tumor antigens. The authors should consider validating their key findings in either MC38 or B16 tumors.
4. As DUSP18 is a phosphatase, transcriptomic profiling might not comprehensively characterize the molecular changes associated with DUSP18 alteration. Proteomic analysis could reveal the impacts of DUSP18 perturbation.
5. For the cytotoxicity assay, the authors used the level of LDH released in culture medium to evaluate the percentage of tumors killed by T cells. However, both tumor and T cells are present in this assay system and both of them can produce LDH. Additional platform should be considered to be used for validation.
6. The conclusion from Figure 5a and 5b would be more convincing, if the authors can show the metabolic changes of shDUSP18 in immunodeficient mice, which could eliminate the

compounding effect from the immune cells to metabolic changes.

7. In the discussion, it would be nice if the authors could provide their perspectives about how DUSP18 is upregulated in tumors.

8. By the end of the paper, the authors listed many other immunoregulator roles of DUSP18, such as regulating MHC class I, mutation burden load. None of these are directly linked to their metabolic mechanisms. How do the authors evaluate the contribution of each role in immune resistance, dispensable or indispensable?

Minor comments

1. In Figure 1p, the N number of the analysis for CD8+ and CD4+ T cells are different. Are these two independent cohorts different? If so, why the authors cannot show the data from the same cohort?

2. There are multiple typos, including but not limited ShREBP2 at page 12 line 234., DUS18 at page 12 line 247

Reviewer #3 (Remarks to the Author): with expertise in cancer immunology, metabolism

In this manuscript, Zhou et al. demonstrated that lanosterol driving through DUSP18-USF1-SREBP2 axis in colo-rectal cancer (CRC) cells inhibits CD8 T cell activation to provide tumor cell immune escape. They further showed that the lanosterol suppresses CD8 T cell mevalonate pathway and downregulates RAS protein prenylation and function, associated with inhibited cytokine production in CD8 T cells. Finally, they combined DUSP18 inhibitor Lumacaftor and anti-PD-1 immunotherapy to demonstrate that blocking DUSP18 improves the infiltration and activation of CD8 T cells in turn to inhibit CRC growth. Overall, they found that DUSP18 dephosphorylates transcription factor USF1 and the stabilized protein can drive Srebp2 gene which in turn accumulates cholesterol biosynthesis.

Taken together, this manuscript provides novel therapeutic alternatives in CRC treatment. We appreciate the thorough investigation of the metabolic pathways but believe that there are some limitations in the immunological part of the story. This study will be largely improved once those concerns can be adequately addressed.

Major comments:

- In Fig. 3, the authors compared tumor growth and weight among USF1 overexpression with different point mutation in USF1 T100. They declared that USF1 is a downstream target of DUSP18 which regulates cholesterol biosynthesis in CRC. However, it is essential to investigate the quality and quantity of CD8 T cell in this setting.

- In Fig. 4a-e, the authors demonstrated that DUSP18-knockdown tumor cells have a higher H2-Kb-SIINFEKL signal. By co-culture assay with tumor cells and activated CD8 T cells to show that DUSP18 knock-down in tumor cells promotes CD8 T cell activation including cytokine production and immune checkpoint inhibitors downregulation. However, the experiment design is weird. Ideally, the tumor cells can induce CD8 T cell activation since pMHC I can be identified. The authors should not pulse tumor cells with OVA peptide in the co-culture assay. Please provide an explanation. It is also necessary to show the CD8 T cell activation markers, which include CD69 and CD25.

- In Fig. 5f and 5h, the authors showed that lanosterol inhibits CD8 T cell proliferation and cytotoxicity. However, the CFSE staining result could not allow us to distinguish between subsequent cell divisions in each group. Please provide a better CD8 T cell proliferation assay. The staining for IFN- γ is also weird. It does not look like the T cells are activated properly in control group.

Minor comments:

- There are some labelling mistakes for gene and protein names throughout the manuscript, which make this manuscript difficult to follow. It requires some revision of the written part.

- The authors performed shRNA for DUSP18 gene knockdown in CRC cell line, shRNA technically produce satisfactory knockdown not knockout. In our understanding the authors mean knockdown but write knockout several times.

Reviewer #4 (Remarks to the Author): with expertise in dual specificity phosphatases, immunology

In this manuscript, Zhou et al. examined the function of DUSP18 in colorectal cancer (CRC). They started with screening of genes that may be important for tumor immune evasion using murine CRC cell line MC38 expressing Cas9 and a mouse CRISPR deletion library against ~1000 genes, which resulted in the identification of DUSP18 as the top candidate gene. The following studies using gene knockdown approaches in both murine and human CRC cell lines and DUSP18 conditional knockout mouse (cKO) CRC models demonstrated that the tumor promoting function of this molecule was associated with reduced CD8⁺ T cell infiltration into the tumors. To understand the mechanism underlying the role of DUSP18 in CRC, RNA-seq analysis of control and DUSP18-knockdown HCT116 cells was carried out. Gene differential expression and gene set enrichment analysis revealed altered cholesterol and fatty acid metabolism pathways in DUSP18-knockdown cells. In addition, USF1, a transcriptional factor required for the expression of SREBP2, was identified as the target of DUSP18. It was found that DUSP18 was able to interact with USF1 to dephosphorylate its Thr100 residue, thereby enhancing its stability and transcriptional activity for regulating the expression of SREBP2. Subsequent analysis of cholesterol synthesis intermediates in DUSP18 knockdown tumor interstitial fluid identified lanosterol as the most significantly down-regulated molecule. Subsequently, it was showed that lanosterol was able to impair CD8⁺ antitumor function. Finally, the authors identified lumacaftor as a potential DUSP18 inhibitor which could inhibit CRC immune evasion. Overall, the authors have performed a large amount of experiments, and the findings, particularly the link between DUSP18 and USF1-SREBP2 is interesting. However, there are multiple issues need to be addressed.

1. Line 156-159 and extended data figure 1g, shRNA-mediated knockdown approach was used to knock down Dusp16 expression in MC38-OVA CRC cells and B16-OVA melanoma cells. Evidence of successful knock down of Dusp16 including both mRNA and protein levels need to be presented here.
2. Line 161-164 and Fig. 1m-o, evidences of CD8⁺ T cell depletion need to be presented.
3. Extended fig 1h-l and Fig. 1p showed that Dusp18 knockdown in MC38-OVA cells didn't change the infiltration of innate immune cells and CD4⁺ T cells, but only resulted in increased CD8⁺ T cell infiltration into the tumors. The numbers of total infiltration cells and each type of cells need to be determined and presented. In addition, the infiltration of NK

cells needs to be examined. Immunohistochemistry analysis of NK cells, CD4+ and CD8+ T cells needs to be presented side-by-side, together with DUSP18 staining. In addition, the flow cytometry data for the quantification of CD8+ TIL in Fig. 1p, 1q and 1r needs to be presented.

4. Fig. 2a-c, WT and Dusp18 cKO mice were used to induce colon tumor formation using the AOM/DSS mouse model. The authors should analyze the tumors from both WT and cKO mice in more details. For examples, the activation of MAPKs, including ERK, JNK and p38 in the tumors, and the expression of cell growth related molecules including c-Myc, Cyclin D1, and Cox-2, etc. need to be evaluated using western blot or immunohistochemistry. In addition, cell proliferation markers such as Ki67 need to be examined by immunohistochemistry.

5. Fig. 2d showed the numbers of CD8+ T cells in tumors from WT and cKO mice. The data showed that quantification was done based on three fields. More fields from more tumors need to be included in the analysis. In addition, other types of cells including CD4+ T cells and NK cells should also be examined.

6. What is the rationale for not using Dusp18 knockdown MC38 cells for the RNA-seq analysis in Fig. 2e? To keep consistency, the RNA-seq analysis should use MC38 cells. In addition, the data presented in Fig. 2e needs to be clarified. For instance, how the comparison was made? Which is the control, the shCtrl cells, or the shDUSP18 cells? In addition, the effect of DUSP18 knockdown in HCT116 cell growth needs to be evaluated both in vitro and in vivo.

7. Fig. 3d showed the regulation of SREBP2 promoter activity by USF1. The data shows that both SREBP2 WT and mutant promoters have increased activity compared to PGL3, and knockdown USF1 resulted in a moderate reduction of luciferase activity, not a dramatic decrease. This suggests that other factors, rather than USF1 play important role in regulation SREBP2 expression.

8. Line 258, "Since the above results indicate that DUSP18's ability to block T cell activation requires its phosphatase activity..", where is the data that support this indication?

9. Fig. 3h, DUSP18 phosphatase dead mutant needs to be included in this experiment.

10. Fig. 4a-b examined the expression of MHC-I molecules in control and DUSP18 knockdown MC38-OVA and B16-OVA cells and found that DUSP18 knockdown cells had higher MHC-I expression compared to control cells. What was the mechanisms underlying

this? Increased cytokine expression in the knockdown cells could be a reason?

11. Fig. 5I-J showed the effect of lanosterol treatment in CD8+ T cells. Lanosterol has been reported to promote tumor cell growth and metastasis. Therefore, reduced lanosterol synthesis could lead to reduced tumor growth. Another question is why reduced lanosterol only resulted in increased CD8+ T cell infiltration, but not CD4+ nor NK cells and other cell types? How this specificity is achieved? These questions need to be addressed with solid experimental data.

12. Line 358-361, "Thus, these results indicate that lanosterol-induced...which in turn inhibits KRAS prenylation, KRAS signaling and CD8+ T cell activation and expansion". This stretched the data (Fig. 5I-n) too far.

13. There are numerous language/grammar errors throughout the manuscript which need to be corrected.

14. The discussion was mainly repeat the results and rather simple.

Point-by-point response to reviewers' comments (NCOMMS-24-07775-T)

We appreciate the thorough constructive suggestions provided by the editor and four reviewers, which have been very helpful in guiding us to further improve our study. With extensive revision, as described in more detail below, we have performed additional experiments to address all the suggestions and comments raised by our Reviewers. Herein, we submit a substantially improved and revised manuscript along with our point-by-point response to each of the Reviewers' point. For their convenience, we have appended all the related figures in this file, which we labeled as Fig. R1 to Fig. R7.

REVIEWER COMMENTS

Reviewer #1 (Remarks to the Author):

In this manuscript, Zhou et al. reported that the DUSP18-USF1-SREBP2-driven transcriptional cascade is involved in CRC immune escape by increasing micro-environmental lanosterol. The CRISPR screening in vivo implicates that depleted of DUSP18 reduced tumor growth rates, which correlate with high levels of CD8+ T cell activation. Mechanistically, DUSP18 dephosphorylates and stabilizes USF1, which induces Srebp2 to accumulate lanosterol and release it into TME. Lanosterol absorbed by CD8+ T cells and reduces RAS protein prenylation and function to inhibits their activation. Finally, the combination of anti-PD-1 antibody and Lumacaftor inhibited CRC growth in mice and synergistically enhanced anti-tumor immunity.

This is a well-performed study. Basically, the data is extensive, detailed and clear. However, the following issues should be addressed:

1. Comment: In the introduction, the authors extensively discussed CTL immune infiltration and cholesterol metabolism, and then mentioned the relationship between phosphatase and immune infiltration. According to the experimental results, the target gene DUSP18 was first screened from the phosphatase CRISPR library, and then the enrichment analysis focused on cholesterol metabolism. Therefore, it is suggested to change the order of presentation.

Response: As suggested, we have changed the order of presentation in the introduction.

2.Comment: The strategy of CRISPR screening in Fig.1a was performed in mice, including Nude mice and C57BL/6J mice. However, it is better to perform genomic and phenotypic analysis of the two strains to identify differences that may influence their rates of tumor growth.

Response: We agree with the Reviewer's suggestion that mouse strain differences might account for differential tumor growth rates. To address this, we reused C57BL/6J-derived *Rag2^{-/-}* mice as the immunodeficient mouse group and wild-type C57BL/6J as the immunocompetent group to perform an unbiased KPD (a Mouse CRISPR Deletion Library) screen. At the time of harvesting 14 days later, tumors grown in immunodeficient hosts were significantly larger (**Fig. R1a-d**). As shown in **Fig. R1e**, similar to the results of the original manuscript screening (**Fig. 1b in the original manuscript**, see also as **Fig. 1b in the revised manuscript**), the sgRNAs targeting *Dusp18* were depleted in tumors engrafted in immunocompetent mice. These results suggest that *Dusp18* is a potentially important regulator of immune evasion in CRC. To further validate the promotional effect of *Dusp18* on tumor immune evasion, we performed subcutaneous transplantation experiments with MC38 mouse CRC cells on *Rag2^{-/-}* mice and WT mice. Silencing of *Dusp18* did not impair tumor growth in immunodeficient *Rag2^{-/-}* mice (**Fig. R1f, g**), whereas it did in immunocompetent C57BL/6J mice (**Fig. R1h, i and Fig. 1h-j in the revised manuscript**). These results suggest a potential role for *Dusp18* in regulating anti-tumor immune responses. We

hope the clarification and additional data could address the suggestions raised by the reviewer.

Fig. R1. In vivo screens with the KPD library uncover novel regulators of immune evasion. **a, b**, Tumor volumes were measured 5 (**a**) and 14 (**b**) days after implantation in the drug targets, kinases and phosphates phosphatases library (KPD) screens. **c, d**, Tumor growth curves (**c**) and weights (**d**) after implantation in the KPD screens. **e**, MAGeCK analysis and RRA ranking of top depleted genes in the screens. Ranked dot plots of depleted genes in immuno-competent hosts compared with immuno-deficient nude mice hosts are shown. **f-g**, 1×10^6 shCtrl or sh*Dusp18* MC38 cells were subcutaneously injected into *Rag2*^{-/-} mice (n = 4). Tumor growth curves (**f**) and tumor weights (**g**) are shown. **h-i**, 1×10^6 shCtrl or sh*Dusp18* MC38 cells were subcutaneously injected into C57BL/6J mice (n = 4). Tumor growth curves (**h**) and tumor weights (**i**) are shown. Data are presented as mean \pm SD (**a-d, f-i**). P-values were calculated unpaired two-tailed t-tests (**a-d**) and one-way ANOVA (**f-i**); ** P < 0.01, *** P < 0.001, **** P < 0.0001, ns, not significant.

3. Comment: In the preliminary screening in Fig.1, it was found that the expression of DUSP18 in CRC tissues was negatively correlated with CTL immune infiltration, indicating that DUSP18 may related with immune suppression in CRC. However, in the animal experiments of Nude mice and C57BL/6J mice, only *Dusp18*-knockdown cells and WT cells were compared, and the experiment should be supplemented that cells with *Dusp18* overexpression were inoculated into mice to observe the CTL infiltration and tumor growth.

Response: To address this suggestion, MC38 cells overexpressing wild-type (WT) and phosphatase dead mutant (Dead) *Dusp18* were inoculated into mice to observe CTL

infiltration and tumor growth. The result showed that only WT *Dusp18* promoted tumor growth in immunocompetent mice and had little effect on the nude mice (**Supplementary Fig. 2c, d in the revised manuscript**). Additionally, only WT *Dusp18* inhibited CD8⁺ T cell infiltration (**Supplementary Fig. 2e, f in the revised manuscript**) and cytotoxic function (**Supplementary Fig. 2g in the revised manuscript**). The exhaustion molecules of CD8⁺ T were also higher only in WT *Dusp18* tumors (**Supplementary Fig. 2h in the revised manuscript**). The above results showed that only *Dusp18* inhibited the CD8⁺ T cell infiltration and CD8⁺ T cell-mediated cytotoxicity, thus indicating that these effects require DUSP18's phosphatase activity. We have included all these newly obtained results in revised manuscript.

4. Comment: In the results of Fig.2, RNA-seq was performed using control and DUSP18-knockdown HCT116 cells to establish the molecular mechanisms of DUSP18 in suppressing anti-tumor immunity. However, it is better to perform the transcriptional sequencing in tumor cells of mice such as MC38 due to the underlying genetic mechanisms of immune suppression may difference in these two strains of cell lines.

Response: Following this suggestion (also suggested by the reviewer #2 and #4), RNA-seq analysis was performed using shCtrl and sh*Dusp18* MC38 cells. GO and KEGG analyses showed that differentially expressed genes were enriched for processes associated with cholesterol biosynthesis and metabolism (**Fig. 2e, f in the revised manuscript**). Heatmaps further demonstrated that genes involved in cholesterol synthesis were significantly downregulated in sh*Dusp18* MC38 cells (**Fig. 2g in the revised manuscript**). GSEA also indicated that genes involved in cholesterol homeostasis were positively enriched in control MC38 cells (**Fig. 2h in the revised**

manuscript). We confirmed decreased mRNA expression and protein levels for selected genes in *shDusp18* MC38 cells and CKO tumor tissues (**Fig. 2i-l and Supplementary Fig. 3a in the revised manuscript**).

5. Comment: In line 210, they hypothesized that DUSP18 regulates a TF that is involved in cholesterol synthesis pathway. However, they found that SREBP2 was responded to DUSP18 knockdown, without any screen strategy. How about other TFs involved in cholesterol synthesis pathway? Do they also respond to DUSP18 knockdown?

Response: We agree that we have not ruled out whether other transcription factors (TFs) that regulate cholesterol synthesis also respond to *Dusp18* inhibition. We therefore examined the expression of several relevant TFs involved in the cholesterol synthesis pathway based on the inhibition of *Dusp18*. These included SREBP1, SREBP2, Ad4BP, Maf, Notch1, XBP1, ROR γ , and c-Myc¹⁻⁸. As shown in **Supplementary Fig. 4b in the revised manuscript**, inhibition of *Dusp18* downregulated only SREBP2 protein levels. This further demonstrates the important role of SREBP2 in the *Dusp18*-mediated cholesterol synthesis pathway.

6. Comment: In Fig.3, they identified that USF1 knockdown declined the expression of SREBP2. It is better to perform the USF1 overexpression experiments.

Response: As suggested, *USF1* was overexpressed in human CRC HCT116 and SW480 and mouse MC38 and B16-OVA cells. Overexpression of *USF1* significantly up-regulated the full (p-SREBP2) and cleaved (n-SREBP2) forms of SREBP2 protein (**Fig.R2**).

Fig. R2. Western blot results in *USF1* overexpression or control tumor cell line.

7. Comment: In Fig. 4, only the Dusp18-Knockdown MC38-OVA model was established, but no Dusp18-overexpression or rescue experiments.

Response: As suggested, we repeated these experiments with the addition of the *Dusp18* rescue experiments (**Fig. 4 in the revised manuscript**).

8. Comment: In Fig. 5e, the authors cultured primary CD8⁺ T cells with different concentrations of exogenous lanosterol. It is better to add an inhibitor to block the endogenous lanosterol as a reverse experiment.

Response: We have now added Ro 48-8071, a Lanosterol Synthase inhibitor, to block the endogenous lanosterol and then cultured primary CD8⁺ T cells with different concentrations of exogenous lanosterol. As shown in **Fig. 5e in the revised manuscript**, Ro 48-8071 treatment enhanced CD69 expression in CD8⁺ T cells, and a gradient of lanosterol was able to decrease CD69 expression in CD8⁺ T cells with or without the addition of Ro 48-8071.

9. Comment: It is suggested that the gene names should be consistent in the same sentence. In the text, when they talking about the same species, the same sentence often mixes human gene like DUSP18 and mouse gene like Dusp18.

Response: In the revised manuscript, the mistake was double-checked and remedied.

10. Comment: It is suggested that the Figures in the text should be arranged according to the order of mention.

Response: The Figures have been rearranged in the text according to the order of mention in the revised manuscript.

11. Comment: The language in the manuscript is acceptable, but further language modification is needed.

Response: The language has been modified throughout the text as appropriate.

Reviewer #2 (Remarks to the Author):

In the manuscript by Zhou X et al. entitled a Colo-Rectal Cancer Immune Escape is Mediated by a DUSP18-USF1-SREBP2-Driven Transcriptional Cascade that Increases Micro-environmental Lanosterol and Blocks CD8+ T Cell Activation, the authors utilized a genetical screen approach to identify the immunological role of tumor intrinsic DUSP18. Silencing Dusp18 in murine colon tumors improves tumor infiltration and effector function of CD8+ T cells. Mechanistically, DUSP18 dephosphorylates Thr100 of USF1, then stabilizes USF1 for transcriptionally activating SREBP2, a gene involved in cholesterol biosynthesis. Ultimately, upregulated DUSP18 in tumors result in accumulation of lanosterol in tumor microenvironment. Increased lanosterol can impair CD8+ T cell tumor infiltration and function. These results reveal a new tumor immune evasion mechanism via the DUSP18-USF1-SREBP2-lanosterol axis. Finally, the authors also identified Lumacaftor as a potent DUSP18 inhibitor by in silico drug screen and demonstrated the potential of Lumacaftor sensitizes colon tumors to immunotherapy. The significance and novelty of this paper are high. The manuscript includes sufficient evidence to support findings based on their own working models and appropriate references to credit previous work. The quality of this manuscript can be further improved from the following aspects. Specifically:

1. Comment: For the CRISPR screen, the authors have chosen the MAGeCK algorithm to analyze their screening results. The assumption of RRA algorithm from MAGeCK is that if a gene has no effect on selection, then sgRNAs targeting that gene should be uniformly distributed across the ranked list of all the sgRNAs. However, the sub-CRISPR library occupies biased distribution during the screening process because of the target gene selection. Therefore, another algorithm should be employed. Also, the authors should provide more information about the genes included in the sub-CRISPR library for better understanding.

Response: We have re-analyzed our CRISPR sequencing data using DrugZ⁹ and DESeq2¹⁰ algorithms. DrugZ analysis compared the gRNA abundance of tumor samples

of nude mice and WT mice, which provided gene-level depletion-normalized Z scores (NormZ) and statistical significance. A more negative normZ score indicates more gRNA depletion in the treated group (WT mice) compared with the control group (Nude mice). As shown in **Supplementary Fig. 1c (left) and Supplementary Table 2 in the revised manuscript**, *Dusp18* has a significant negative normZ score, suggesting that there is more gRNA depletion in the WT mice and that *Dusp18* is able to promote immune evasion of tumors. As shown in **Fig. 1c (right) and Supplementary Table 2 in the revised manuscript** that DESeq2 analysis yielded similar results, i.e., the gRNA abundance of *Dusp18* was significantly downregulated in WT mouse tumors. In summary, we employed a total of three algorithms for CRISPR analysis, all of which consistently identified *Dusp18* as a potential and important regulator of immune evasion in colorectal cancer. Meanwhile, we provided relevant information about the genes included in the sub-CRISPR library, as shown in **Supplementary Table S1** in the revised manuscript.

2. Comment: The authors used two lines from different species to explore the role of DUSP18. The identification of DUSP18 was used a murine line, but the mechanistic studies in Fig 3 were used the human line, HCT116. At least, the authors should consider confirming the identified transcriptional changes in MC38 tumors.

Response: Please see the response to Comment 4 of the Reviewer #1 (Page 4).

3. Comment: Their preclinical findings related to antitumor immune responses are largely dependent on OVA-expressing tumor cells. As OVA is a foreign antigen, these models might not well represent immune responses against tumor antigens. The authors should consider validating their key findings in either MC38 or B16 tumors.

Response: We have provided more information and performed the following experiments with MC38 tumors to address the suggestions. To detect the anti-tumor immune response triggered by the tumor antigen itself, we validated our key experimental results using MC38 cells (not expressing OVA). Our new findings show the following:

(a) Silencing of *Dusp18* did not impair tumor growth in immunodeficient mice (**Fig. 1f, g in the revised manuscript**). However, in immunocompetent mice, *Dusp18* downregulation significantly decreased tumor growth (**Fig. 1h-j in the revised manuscript**), which is consistent with our original manuscript. These results show that adaptive immune response plays an important role in *Dusp18*-mediated tumor growth.

(b) In immunocompetent C57BL/6J mice, inhibition of *Dusp18* significantly decreased tumor growth and increased CD8⁺ T cell infiltration (**Fig. 1n and Supplementary Fig. 2a in the revised manuscript**). It also allowed higher expression levels of the cytotoxic molecules IFN- γ and granzyme B (**Fig. 1o in the revised manuscript**), and lower expression of the inhibitory PD-1, TIM-3 and CTLA-4 (**Fig. 1p and Supplementary Fig. 2b in the revised manuscript**). These results indicate that inhibition of *Dusp18* enhances the cytotoxic activity of CD8⁺ T cells to mediate tumor killing.

(c) MC38 cells overexpressing WT and phosphatase dead *Dusp18* were inoculated into mice to observe the CTL infiltration and tumor growth. Only the former cells promoted tumor growth in immunocompetent mice but not in nude mice (**Supplementary Fig. 2c, d in the revised manuscript**). WT *Dusp18* overexpression also inhibited CD8⁺ T cell infiltration (**Supplementary Fig. 2e, f in the revised manuscript**) and cytotoxic function

(**Supplementary Fig. 2g in the revised manuscript**). The exhaustion molecules in CD8⁺ T were also higher in WT *Dusp18* tumors (**Supplementary Fig. 2h in the revised manuscript**). Thus, the above results showed that WT *Dusp18* (but not phosphatase dead one) could inhibit the CD8⁺ T cell infiltration and activation and suppress CD8⁺ T cell-mediated cytotoxicity.

(**d**) The syngeneic mouse MC38 CRC model was utilized to examine how Lumacaftor, with or without the addition of anti-PD-1 antibody, affected tumor growth and survival. Strikingly, combination therapy significantly suppressed tumor growth and prolonged survival of MC38 tumor-bearing immunocompetent C57BL/6 mice compared to either single-agent or control-treated group (**Fig. 8a-c in the revised manuscript**). Analysis of tumor-associated immune cells demonstrated that combination Lumacaftor plus anti-PD-1 treatment significantly increased the percentage of CD8⁺ T cells but had no significant effect on the proportion of CD4⁺ cells (**Fig. 8d, e in the revised manuscript**). Combined treatment also significantly elevated the expression of GzmB and IFN- γ by these CD8⁺ T cells (**Fig. 8f, g in the revised manuscript**).

4. Comment: As DUSP18 is a phosphatase, transcriptomic profiling might not comprehensively characterize the molecular changes associated with DUSP18 alteration. Proteomic analysis could reveal the impacts of DUSP18 perturbation.

Response: To address the Reviewer's question, proteomic analysis has now been performed. As shown in **Supplementary Fig. 4a in the revised manuscript**, inhibition of *Dusp18* reduced the expression of 440 proteins and increased the expression of 449 proteins. KEGG signaling pathway enrichment showed that proteins as a result of

Dusp18 inhibition were mainly enriched in metabolic pathways and cholesterol biosynthesis signals (**Supplementary Fig. 4b in the revised manuscript**). Proteins up-regulated by *Dusp18* inhibition were mainly enriched in antigen presentation signals (**Supplementary Fig. 4c in the revised manuscript**). These results were consistent with our RNA-Seq findings. Notably inhibition of *Dusp18* decreased important enzymes associated with cholesterol biosynthesis, including SREBP2, HMGCR, LSS, and SQLE et al (**Fig. 2m**). The above results indicate the importance and conservation of DUSP18 in regulating the cholesterol synthesis pathway.

5. Comment: For the cytotoxicity assay, the authors used the level of LDH released in culture medium to evaluate the percentage of tumors killed by T cells. However, both tumor and T cells are present in this assay system and both of them can produce LDH. Additional platform should be considered to be used for validation.

Response: Responding to this suggestion, we have performed additional cytotoxicity experiments using MC38/B16-OVA-Luc cell lines (stable overexpression of ovalbumin and luciferase). The viability of the tumor cells was positively correlated with the fluorescent signal. Since CD8⁺ T cells do not express luciferase, their contribution can be excluded. As shown in **Fig. 4f, g in the revised manuscript**, inhibition of *Dusp18* significantly enhanced tumor cell killing by CD8⁺ T cells.

6. Comment: The conclusion from Figure 5a and 5b would be more convincing, if the authors can show the metabolic changes of shDUSP18 in immunodeficient mice, which could eliminate the compounding effect from the immune cells to metabolic changes.

Response: As suggested, we have now used cholesterol metabolomics to determine the levels of cholesterol biosynthetic intermediates in the tumor interstitial fluid of

subcutaneous tumors from *shDusp18* and *shCtrl* MC38 cells in immunodeficient mice. The levels of lanosterol, numerous cholesterol synthesis intermediates, oxysterols and other derivatives were significantly lower in the fluid from *shDusp18* tumors (Fig. R3a, b), which is consistent with the results in our original manuscript. By performing our studies in immuno-deficient mice, we were able to eliminate any contribution from the immune cell populations.

Fig. R3. Metabolomics of cholesterol metabolism in tumor interstitial fluids from nude mice. a, Cholesterol metabolism-related metabolites in interstitial fluids from subcutaneous *shCtrl* and *shDusp18* MC38 tumors in Nude mice. **b,** Volcano plot showing differences in lanosterol and other metabolites in *shCtrl* and *shDusp18* MC38 tumors.

7. Comment: In the discussion, it would be nice if the authors could provide their perspectives about how DUSP18 is upregulated in tumors.

Response: As suggested, we have now discussed the most likely reasons for the upregulation of DUSP18 in human tumors.

8. Comment: By the end of the paper, the authors listed many other immunoregulator roles of DUSP18, such as regulating MHC class I, mutation burden load. None of these

are directly linked to their metabolic mechanisms. How do the authors evaluate the contribution of each role in immune resistance, dispensable or indispensable?

Response: The recognition and killing of tumor cells by the immune system requires a series of processes known as the cancer-immunity cycle (CI cycle)¹¹. This involves the release of tumor antigens, antigen presentation, immune cell activation and immune infiltration. In our original manuscript, we analyzed the TCGA-CRC data and found that *DUSP18* mRNA expression significantly negatively correlated with the antigen presentation process (**Supplementary Fig. 9c and 10b, d, e in the revised manuscript**), which was also verified by flow cytometry (**Fig. 4a-d in the revised manuscript**). We also found that *DUSP18* mRNA expression negatively correlated with tumor antigenic mutational load (**Supplementary Fig. 9e in the revised manuscript**). However, we believe that *DUSP18* could regulate multiple biological processes in the CI cycle, thus allowing individual processes to act synergistically to further better promote tumor immune evasion. Therefore, we believe that the contribution of each role including MHC class I and mutation burden load in immune resistance was dispensable, and that each process acts independently but synergistically to exert anti-tumor effects. To test this conjecture, we first performed the following experiments on the process MHC-I. We used MC38-OVA-Luc cells and OT-1 CD8⁺ T cells (CD8⁺ T cells were separated from OT1-C57BL/6 mice spleen and pre-activated by anti-CD3 and anti-CD28 antibodies) for tumor cell-killing assay (**Fig. R4a**). Inhibition of *Dusp18* significantly enhanced CTL cytotoxicity against the target tumor cells (**Fig. R4b**), which is consistent with the results in our original manuscript (**Fig. 4f, g in the revised manuscript**). However, treatment with MHC-I antibody (anti-H2-Kb-SIINFEKL) did not completely eliminate

the difference for T cells eliciting cytotoxic effects between shCtrl and sh*Dusp18* cells (**Fig. R4b**). This supported the idea that the increased antigen-specific T cell killing elicited by inhibition of *Dusp18* was partly due to increased MHC-I expression.

To examine the role of mutation burden load in immune resistance, we analyzed subcutaneous CT26 tumor growth in BALB/c mice. CT26 cells are considered to be a low immunogenicity murine CRC line. To increase tumor mutation burden and higher levels of predicted neoantigens in CT26 cells, we knocked out the *MLH1* mismatch repair gene to artificially cause mismatch repair defects, which then lead to microsatellite high instability (MSI-H) and a greater mutation load¹² (**Fig.R4c**). *Dusp18* overexpression significantly promoted tumor proliferation in the sgCtrl group (**Fig.R4d**), which is consistent with the results in our revised manuscript (**Supplementary Fig. 2d in the revised manuscript**). Despite the fact that *Mlh1* knockout inhibited the tumor growth, *Dusp18* overexpression still promoted tumor growth (**Fig.R4d**). This suggested that *Dusp18*'s ability to promote tumor immune evasion does not depend only on its suppression of tumor mutational load. For the above phenomenon, we believe that inhibition of *Dusp18* should also allow for the activation of CD8⁺ T cells through a metabolic mechanism that reduces lanosterol release. Thus, inhibition or overexpression *Dusp18* can still affect the killing of tumor cells by CD8⁺ T cells after the addition of MHC-I antibody or the *Mlh1* knockout. These results further demonstrate that *DUSP18* synergistically promotes CRC immune evasion via multiple mechanisms. It was further shown that *DUSP18* plays a crucial role in the immune evasion.

Fig. R4. The effect of *Dusp18* on MHC-I and mutation burden load in tumor immune evasion. **a**, Schematic representation of an in vitro T cell killing assay. **b**, MC38-OVA-Luc shCtrl and sh*Dusp18* cells were cocultured with CD8⁺ T cells isolated from OVA-specific T cell receptor transgenic (OT1) mice. The percent lysis of shCtrl and sh*Dusp18* tumor cells after incubation with OT-1 CTLs (n = 3 biological replicates). Effector to target (E/T) ratios are shown. **c**, Western blot analysis of MLH1 protein expression of the sgCtrl and sg*Mlh1* CT26 tumor cell lines. **d**, Tumor growth and tumor weight in BALB/c mice subcutaneously injected with indicated CT26 cells. Data are presented as mean ± SD (**b, d**). *P*-values were calculated unpaired two-tailed t-tests (**b**) and one-way ANOVA (**d**); ***P* < 0.01, ****P* < 0.001, *****P* < 0.0001.

Minor comments

1. Comment: In Figure 1p, the N number of the analysis for CD8⁺ and CD4⁺ T cells are different. Are these two independent cohorts different? If so, why the authors cannot show the data from the same cohort?

Response: In fact, the CD8⁺ and CD4⁺ T cells analyzed in **Figure 1p in the original manuscript** represent two different cohorts. We agree with the reviewer that the same cohort should have been provided to explore this issue. We have thus re-examined the frequencies in CD8⁺ and CD4⁺ T cells from the same cohort from MC38 tumors (**Fig. 1n in the revised manuscript**). In addition, for all other related experiments, we show the data from the same cohort and we have included all these newly obtained results in revised manuscript.

2. Comment: There are multiple typos, including but not limited ShREBP2 at page 12 line 234., DUS18 at page 12 line 247.

Response: These typos were double-checked and fixed in the revised manuscript.

Reviewer #3 (Remarks to the Author):

In this manuscript, Zhou et al. demonstrated that lanosterol driving through DUSP18-USF1-SREBP2 axis in colo-rectal cancer (CRC) cells inhibits CD8 T cell activation to provide tumor cell immune escape. They further showed that the lanosterol suppresses CD8 T cell mevalonate pathway and downregulates RAS protein prenylation and function, associated with inhibited cytokine production in CD8 T cells. Finally, they combined DUSP18 inhibitor Lumacaftor and anti-PD-1 immunotherapy to demonstrate that blocking DUSP18 improves the infiltration and activation of CD8 T cells in turn to inhibit CRC growth. Overall, they found that DUSP18 dephosphorylates transcription factor USF1 and the stabilized protein can drive Srebp2 gene which in turn accumulates cholesterol biosynthesis.

Taken together, this manuscript provides novel therapeutic alternatives in CRC treatment. We appreciate the thorough investigation of the metabolic pathways but believe that there are some limitations in the immunological part of the story. This study will be largely improved once those suggestions can be adequately addressed.

Major comments:

1. Comment: In Fig. 3, the authors compared tumor growth and weight among USF1 overexpression with different point mutation in USF1 T100. They declared that USF1 is a downstream target of DUSP18 which regulates cholesterol biosynthetic in CRC. However, it is essential to investigate the quality and quantity of CD8 T cell in this setting.

Response: Following the reviewer's suggestion, we have investigated the quality and quantity of CD8⁺ T cell in these experiments (**Fig. 3m-o in the revised manuscript**).

2. Comment: In Fig. 4a-e, the authors demonstrated that DUSP18-knockdown tumor cells have a higher H2-Kb-SIINFEKL signal. By co-culture assay with tumor cells and activated CD8 T cells to show that DUSP18 knock-down in tumor cells promotes CD8 T cell activation including cytokine production and immune checkpoint inhibitors downregulation. However, the experiment design is weird. Ideally, the tumor cells can induce CD8 T cell activation since pMHCI can be identified. The authors should not pulse tumor cells with OVA peptide in the co-culture assay. Please provide an explanation. It is also necessary to show the CD8 T cell activation markers, which include CD69 and CD25.

Response: We apologize for any misunderstanding caused by the lack of sufficient detail for the co-culture experiments in the original manuscript. In these original

studies, we added OVA peptide to enhance CD8⁺ T cell-mediated killing during in vitro killing experiments. In the co-culture assays, we directly co-cultured MC38 and CD8⁺ T cells without adding OVA peptide in the original manuscript. This is indeed a misunderstanding on the part of the reviewers due to a writing error in our methodology in the original manuscript. A clearer description of these experiments has been added to the revised manuscript.

In the revised manuscript, for the co-culture assay (**Fig. 4h in the revised manuscript**), we used sh*Dusp18* or shCtrl MC38 tumor cells (not expressing Ovalbumin; not pulse OVA peptide) co-cultured with CD8⁺ T cells, which were subsequently assayed for CD8⁺ T cell-associated markers, cytokine production and immune checkpoint related proteins expression (**Fig. 4i-l in the revised manuscript**). We have noted and now rectified the problem. However, we used MC38-OVA cells (MC38 cells stably overexpressing Ovalbumin; not pulse OVA peptide) to detect changes in antigen-presenting molecules (**Fig. 4a-d in the revised manuscript**), as well as in CD8⁺ T cell-mediated tumor cell-killing assays using MC38-OVA-Luc cells (**Fig. 4e in the revised manuscript**) and OT-1 CD8⁺ T cells (CD8⁺ T cells were separated from OT1-C57BL/6 mice spleen and pre-activated by anti-CD3 and anti-CD28 antibodies) (**Fig. 4e, f in the revised manuscript**). Following our reviewer's suggestion, CD8⁺ T cell activation markers, including CD69 and CD25, were detected in co-culture experiments (**Fig. 4i in the revised manuscript**). And we have included all these newly obtained results in revised manuscript.

3. Comment: In Fig. 5f and 5h, the authors showed that lanosterol inhibits CD8 T cell proliferation and cytotoxicity. However, the CFSE staining result could not allow us to distinguish between subsequent cell divisions in each group. Please provide a better CD8 T cell proliferation assay. The staining for IFN- γ is also weird. It does not look like the T cells are activated properly in control group.

Response: We agree with the reviewer's opinion that we should provide a better CD8⁺ T cell proliferation assay. To this end, we used EdU (5-ethynyl-2-deoxyuridine), a thymidine analog, that is incorporated into newly synthesized DNA and can be used as a proliferation marker. As shown in **Fig. 5f in the revised manuscript**, lanosterol treatment significantly inhibited CD8⁺ T cell proliferation when measured by this method. Regarding the staining for IFN- γ in CD8⁺ T cells, we have repeated this assay and used the unstained CD8⁺ T cells as negative control to gate the area and confirmed that CD8⁺ T cells are activated properly in control group (**Fig.R5 and Fig. 5h in the revised manuscript**).

Fig.R5. The staining for IFN- γ in CD8⁺ T cells with control and lanosterol treatment.

Minor comments:

1. Comment: There are some labelling mistakes for gene and protein names throughout the manuscript, which make this manuscript difficult to follow. It requires some revision of the written part.

Response: We have revised the manuscript to ensure accuracy and consistency of gene and protein names.

2. Comment: The authors performed shRNA for DUSP18 gene knockdown in CRC cell line, shRNA technically produce satisfactory knockdown not knockout. In our understanding the authors mean knockdown but write knockout several times.

Response: Following our reviewer's suggestion, we identified and corrected these errors in the revised manuscript.

Reviewer #4 (Remarks to the Author):

In this manuscript, Zhou et al. examined the function of DUSP18 in colorectal cancer (CRC). They started with screening of genes that may be important for tumor immune evasion using murine CRC cell line MC38 expressing Cas9 and a mouse CRISPR deletion library against ~1000 genes, which resulted in the identification of DUSP18 as the top candidate gene. The following studies using gene knockdown approaches in both murine and human CRC cell lines and DUSP18 conditional knockout mouse (cKO) CRC models demonstrated that the tumor promoting function of this molecule was associated with reduced CD8+ T cell infiltration into the tumors. To understand the mechanism underlying the role of DUSP18 in CRC, RNA-seq analysis of control and DUSP18-knockdown HCT116 cells was carried out. Gene differential expression and gene set enrichment analysis revealed altered cholesterol and fatty acid metabolism pathways in DUSP18-knockdown cells. In addition, USF1, a transcriptional factor required for the expression of SREBP2, was identified as the target of DUSP18. It was found that DUSP18 was able to interact with USF1 to dephosphorylate its Thr100 residue, thereby enhancing its stability and transcriptional activity for regulating the expression of SREBP2. Subsequent analysis of cholesterol synthesis intermediates in DUSP18 knockdown tumor interstitial fluid identified lanosterol as the most significantly down-regulated molecule. Subsequently, it was showed that lanosterol was able to impair CD8+ antitumor function. Finally, the authors identified lumacaftor as a potential DUSP18 inhibitor which could inhibit CRC immune evasion. Overall, the authors have performed a large amount of experiments, and the findings, particularly the link between DUSP18 and USF1-SREBP2 is interesting. However, there are multiple issues need to be addressed.

1. Comment: Line 156-159 and extended data figure 1g, shRNA-mediated knockdown approach was used to knock down Dusp18 expression in MC38-OVA CRC cells and B16-OVA melanoma cells. Evidence of successful knock down of Dusp18 including both mRNA and protein levels need to be presented here.

Response: Evidence of successful inhibition of *Dusp18* including both mRNA (**Supplementary Fig. 1h in the revised manuscript**) and protein (**Supplementary Fig. 1g in the revised manuscript**) levels in MC38 and B16-OVA cells are presented in the revised manuscript.

2. Comment: Line 161-164 and Fig. 1m-o, evidences of CD8⁺ T cell depletion need to be presented.

Response: Evidence for CD8⁺ T cell depletion is presented in **Supplementary Fig. 1n in the revised manuscript**.

3. Comment: Extended fig 1h-I and Fig. 1p showed that *Dusp18* knockdown in MC38-OVA cells did not change the infiltration of innate immune cells and CD4⁺ T cells, but only resulted in increased CD8⁺ T cell infiltration into the tumors. The numbers of total infiltration cells and each type of cells need to be determined and presented. In addition, the infiltration of NK cells needs to be examined. Immunohistochemistry analysis of NK cells, CD4⁺ and CD8⁺ T cells needs be presented side-by-side, together with DUSP18 staining. In addition, the flow cytometry data for the quantification of CD8⁺ TIL in Fig. 1p, 1q and 1r needs to be presented.

Response: Following the reviewer's suggestion, we have determined and presented the numbers of total infiltration cells and each type of cell numbers including NK cells (**Supplementary Fig. 1o, p in the revised manuscript**). Immunohistochemistry analysis of NK cells, CD4⁺ and CD8⁺ T cells have been presented side-by-side, together with *Dusp18* staining in the **Supplementary Fig. 2a of the revised manuscript**. The flow cytometry data for the quantification of CD8⁺ TIL in Fig. 1p, 1q and 1r in the original manuscript have been presented in the **Fig. 1n, o and Supplementary Fig. 2b in the revised manuscript**.

4. Comment: Fig. 2a-c, WT and *Dusp18* cKO mice were used to induce colon tumor formation using the AOM/DSS mouse model. The authors should analyze the tumors from both WT and cKO mice in more details. For examples, the activation of MAPKs, including ERK, JNK and p38 in the tumors, and the expression of cell growth related molecules including c-Myc, Cyclin D1, and Cox-2, etc. need to be evaluated using western blot or immunohistochemistry. In addition, cell proliferation markers such as Ki67 need to be examined by immunohistochemistry.

Response: We analyzed tumors from both *Dusp18* WT and conditional KO (CKO) mice for evidence of ERK, JNK and p38 activation and the expression of some cell growth related molecules by western blot. We found that tumors from *Dusp18* CKO mice has higher levels of JNK phosphorylation (Phospho-JNK Tyr185), while ERK and p38 remained almost unchanged among several MAPKs (**Supplementary Fig. 2k in the revised manuscript**). Some cell growth-related molecules (c-Myc, Cyclin D1, Cox-2 and Ki-67) were not significantly different in these two tumor tissues (**Supplementary Fig. 2k, j in the revised manuscript**).

5. Comment: Fig. 2d showed the numbers of CD8⁺ T cells in tumors from WT and cKO mice. The data showed that quantification was done based on three fields. More fields from more tumors need to be included in the analysis. In addition, other types of cells including CD4⁺ T cells and NK cells should also be examined.

Response: Following the reviewers' suggestion, we analyzed six tumor samples per group, each with five randomly selected fields of view to quantify tumor-infiltrating CD8⁺ T cells, CD4⁺ T cells and NK cells (**Fig. 2d and Supplementary Fig. 2j in the revised manuscript**). Our results showed there to be no significant differences between these two types of cells in WT and CKO tumor groups.

6. Comment: What is the rationale for not using *Dusp18* knockdown MC38 cells for the RNA-seq analysis in Fig. 2e? To keep consistency, the RNA-seq analysis should use MC38 cells. In addition, the data presented in Fig. 2e needs to be clarified. For instance,

how the comparison was made? Which is the control, the shCtrl cells, or the shDUSP18 cells? In addition, the effect of DUSP18 knockdown in HCT116 cell growth needs to be evaluated both in vitro and in vivo.

Response: We initially identified *Dusp18* as a novel key gene regulating tumor immunity by CRISPR library screening using mouse MC38 cells. We then validated its promotion of CD8⁺ T cell-dependent tumor immune evasion. Following this, we performed a clinical correlation analysis of *DUSP18* mRNA level in clinical samples before proceeding to downstream mechanistic analyses. As to why we chose HCT116 instead of MC38 cells, there is a large amount of literature based on using mouse cells for mechanistic studies, which may be due to some biological differences between mice and human. However, we strongly agree with the reviewer that mouse MC38 CRC cells should be used for RNA-seq to maintain the consistency of the study and to validate the findings of the original manuscript. Therefore, RNA-seq analysis was performed using shCtrl and sh*Dusp18* MC38 cells in the revised manuscript. Please see the response to Comment 4 of the Reviewer #1 (Page 4).

We are sorry for missing detailed description of the data presented in **Fig. 2e in the original manuscript** (see also as **Supplementary Fig. 3b in the revised manuscript**). The GSEA analysis was also performed based on the shCtrl versus sh*DUSP18* HCT116 cells. shCtrl cells group is the control, the sh*DUSP18* cells group is the treatment. In the revised manuscript, we have added more description of the figures and how experiments were done in both the main text and the section of method.

As suggested, the effect of *DUSP18* inhibition in HCT116 cell growth was evaluated both in vitro (**Supplementary Fig. 1k in the revised manuscript**) and in vivo (**Supplementary Fig. 1l in the revised manuscript**).

7. Comment: Fig. 3d showed the regulation of SREBP2 promoter activity by USF1. The data shows that both SREBP2 WT and mutant promoters have increased activity compared to PGL3, and knockdown USF1 resulted in a moderate reduction of luciferase activity, not a dramatic decrease. This suggests that other factors, rather than USF1 play important role in regulation SREBP2 expression.

Response: We have increased the experimental sample numbers and repeated the experiment discussed above (**Fig. 3d in the revised manuscript**). This showed that shRNA-mediated *USF1* inhibition dramatically decreased luciferase activity driven by *SREBF2* WT promoters, but not when the promoter contained mutant *USF1* binding sites. Furthermore, *USF1* inhibition significantly reduced the *SREBF2* mRNA level and protein abundance (**Fig. 3b and Supplementary Fig. 5c in the revised manuscript**), and *DUSP18* overexpression was unable to upregulate SREBP2 protein in *USF1* inhibition cells (**Fig. 3k in the revised manuscript**). These results suggest that *USF1* plays an indispensable role in the regulation of SREBP2 by *DUSP18*.

8. Comment: Line 258, Since the above results indicate that *DUSP18*'s ability to block T cell activation requires its phosphatase activity, where is the data that support this indication?

Response: To address this, MC38 cells overexpressing WT and phosphatase dead mutant *Dusp18* were inoculated into mice to observe the CTL infiltration and tumor growth. WT MC38 cells but not the Dead MC38 cells promoted tumor growth in immunocompetent mice and showed no growth differences in immuno-deficient mice

(**Supplementary Fig. 2c, d in the revised manuscript**). Only WT *Dusp18* overexpression and not dead one inhibited CD8⁺ T cell infiltration (**Supplementary Fig. 2e, f in the revised manuscript**) and cytotoxic function (**Supplementary Fig. 2g in the revised manuscript**). The exhaustion molecules in CD8⁺ T were also higher in WT *Dusp18* overexpression tumors (**Supplementary Fig. 2h in the revised manuscript**). Thus, the above results showed that the ability of *Dusp18* to inhibit the CD8⁺ T cell infiltration and CD8⁺ T cell-mediated cytotoxicity is dependent on its phosphatase activity.

9. Comment: Fig. 3h, DUSP18 phosphatase dead mutant needs to be included in this experiment.

Response: We fully agree with the reviewer that results with DUSP18 phosphatase dead mutant should be included. To this end, we repeated this experiment and added the DUSP18 phosphatase dead mutant in this experiment (**Fig. 3h in the revised manuscript**).

10. Comment: Fig. 4a-b examined the expression of MHC-I molecules in control and DUSP18 knockdown MC38-OVA and B16-OVA cells and found that DUSP18 knockdown cells had higher MHC-I expression compared to control cells. What was the mechanisms underlying this? Increased cytokine expression in the knockdown cells could be a reason?

Response: To address the reviewer's suggestion, we first reviewed the signals regulated by DUSP18 and then found that inhibition of *DUSP18* significantly enhanced interferon signaling (**Fig. 2e in the original manuscript**), which is known to be important in regulating antigen presentation. We also performed GO and GSEA analyses of genes upregulated after inhibition of *DUSP18* and found similar results (**Fig. R6a, b**). To further validate this finding in other datasets, we grouped the TCGA-CRC

data into high and low expression groups based on the median *DUSP18* mRNA levels and found that *DUSP18*-low samples were more enriched in antigen presentation and interferon response signaling (**Supplementary Fig. 10b, d and e in the revised manuscript**). Based on the above findings, we also agree with the Reviewer that inhibition of *DUSP18* upregulates interferon and cytokine production and promotes antigen presentation signaling. As shown in **Fig. R6c**, interferon-gamma (IFN- β) treatment significantly increased *Dusp18* inhibition-mediated MHC-I expression. Therefore, increased cytokine expression in the *shDUSP18* cells could be an important reason for the MHC-I up-regulation.

Fig. R6. Inhibition of *Dusp18* induces MHC-I expression. **a**, GO analysis of genes upregulated after inhibition of *DUSP18*. **b**, GSEA analysis on interferon response signaling. **c**, Expression levels of MHC-I of *shDusp18* and *shCtrl* MC38-OVA in the presence or absence of IFN- β treatment were determined by FACS. (MFI, mean fluorescence intensity).

11. Comment: Fig. 5I-J showed the effect of lanosterol treatment in CD8 $^+$ T cells. Lanosterol has been reported to promote tumor cell growth and metastasis. Therefore, reduced lanosterol synthesis could lead to reduced tumor growth. Another question is why reduced lanosterol only resulted in increased CD8 $^+$ T cell infiltration, but not CD4 $^+$ nor NK cells and other cell types? How this specificity is achieved? These questions need to be addressed with solid experimental data.

Response: We appreciate this suggestion regarding the role of lanosterol in tumor proliferation and CD8 $^+$ T cells specificity. Indeed, it has been reported that inhibition of lanosterol production can inhibit the proliferation of hepatocellular carcinoma¹³,

glioma^{14,15}, and pancreatic cancer tumors¹⁶ as well as the metastatic ability of CRC cells¹⁶. However, it also has been reported that dietary lanosterol can suppress aberrant colonic crypt formation¹⁷ and inhibit hormone-dependent growth of breast cancer cells^{18,19}. Due to the heterogeneity of different tumors, the mechanism of action of lanosterol on tumors is likely to vary and be context-dependent.

The essentiality of lanosterol for tumor growth reflects its dependence and the amount it is able to synthesize. When production is high, the inhibition of cholesterol or lanosterol production will almost certainly inhibit their growth. For example, over-expressing oncogenes such as *Myc* or inhibiting tumor suppressor genes such as *p53* may be associated with a large requirement for cholesterol synthesis, under which lanosterol becomes indispensable (**Fig. R7a**). In other cases, however, inhibition of lanosterol synthesis may not significantly affect cell viability per se, possibly because the requirement is relatively low or other compensatory pathways are activated (**Fig. R7a**). In CRC, asynchronous alternation of enzymes along the pathway may occur. For example, the mRNA levels of *HMGCR*, *DHCR24* and *IDI1* did not differ between tumor and normal tissues (**Fig. R7b**), thus initially conveying the impression that CRC is not overly dependent on the cholesterol synthesis pathway. However, the mRNA levels of *SQLE* and *LSS*, both of which are key enzymes for the synthesis of lanosterol and cholesterol, were significantly up-regulated (**Fig. R7c**). Such an asynchronous pattern of up-regulation could theoretically lead to the accumulation of certain critical intermediates such as lanosterol, which, in the context of CRC, are more important for immune suppression than for more traditional functions such as tumor growth. Such

an interpretation could explain why *DUSP18* inhibition-mediated reduction of lanosterol and its downstream products did not directly affect proliferation. In such a scenario, the importance of lanosterol would be highlighted only in an immunocompetent system where it could potentially be mis-interpreted as being necessary for proliferation.

To address the question of lanosterol's specific effects on CD8⁺ T cells, we conducted the following experiments. Given that lanosterol inhibits CD8⁺ T cells by decreasing the protein level of HMGCR in CD8⁺ T cells, we analyzed its expression in different immune cells in three single-cell datasets. In all 3 data sets (GSE139555, GSE146771 and GSE178341), the mRNA level of *HMGCR* was significantly higher in CD8⁺ T cells than in CD4 T cells, NK cells and monocytes, among others (**Fig. R7d- R7f**). These results suggest that the expression level of *HMGCR* in CD8⁺ T cells is much higher than that in most other immune cells, which seems to indicate that CD8⁺ T cells were highly dependent on HMGCR, which plays a crucial role in their activation or function. This may be a key reason for the more potent inhibitory effect of lanosterol on CD8⁺ T cells. To further validate this finding, we extracted primary CD8⁺ T cells, CD4⁺ T cells, and NK cells from mice for RT-qPCR validation of *HMGCR* gene expression. The result revealed that *HMGCR* was highly expressed only in CD8⁺ T cells (**Fig. R7g**), suggesting that CD8⁺ T cells engage in a more active program of HMGCR-mediated mevalonate pathway.

Fig. R7. Multiple genes expression and distribution patterns in human CRC. **a**, Effect of Ro 48-8071 on proliferation of HCT116 *p53* WT or *p53*^{-/-} cells. **b**, **c**, Transcriptional levels of indicated genes in CRC tumors and normal tissues from TCGA-COAD. **d-f**, Analysis for *HMGCR* mRNA expression of immune cell population in colorectal cancer patients by using the published single-cell RNA-seq data GSE139555 (**d**), GSE146771 (**e**), GSE178341 (**f**). **g**, *HMGCR* mRNA expression of indicated immune cells. Data are presented as mean ± SD (**a-g**). *P*-values were calculated unpaired two-tailed t-tests (**a-c**) and one-way ANOVA (**g**); ** *P* < 0.01, *** *P* < 0.001, **** *P* < 0.0001.

12. Comment: Line 358-361, Thus, these results indicate that lanosterol-induced which in turn inhibits KRAS prenylation, KRAS signaling and CD8+ T cell activation and expansion. This stretched the data (Fig. 5I-n) too far.

Response: To address this deficiency, we have modified the presentation of the relevant results in the revised manuscript (P20, Line 405-408).

13. Comment: There are numerous language/grammar errors throughout the manuscript which needs to be corrected.

Response: We have scrutinized and revised the language/grammar errors in the revised manuscript.

14. Comment: The discussion was mainly repeat the results and rather simple.

Response: The discussion section has been rewritten so as to now further analyze and explain the significance and possible implications of the findings in more depth. The revised discussion section will be more substantial and in-depth to meet the expectations of reviewers and readers.

References:

- 1) Luo, J., Yang, H. & Song, B. L. Mechanisms and regulation of cholesterol homeostasis. *Nat Rev Mol Cell Biol* **21**, 225-245, (2020).
- 2) Tumor-secreted FGF21 acts as an immune suppressor by rewiring cholesterol metabolism of CD8+T cells. *Cell Metabolism*.
- 3) Baba, T. *et al.* Ad4BP/SF-1 regulates cholesterol synthesis to boost the production of steroids. *Commun Biol* **1**, 18, (2018).
- 4) Kim, M. *et al.* Maf links Neuregulin1 signaling to cholesterol synthesis in myelinating Schwann cells. *Genes Dev* **32**, 645-657, (2018).
- 5) Yang, Y. *et al.* Osteoblasts impair cholesterol synthesis in chondrocytes via Notch1 signalling. *Cell Proliferation* **54**, (2021).
- 6) Yang, Z. *et al.* Cancer cell-intrinsic XBP1 drives immunosuppressive reprogramming of intratumoral myeloid cells by promoting cholesterol production. *Cell Metab* **34**, 2018-2035 e2018, (2022).
- 7) Cai, D. *et al.* ROR γ is a targetable master regulator of cholesterol biosynthesis in a cancer subtype. *Nature Communications* **10**, (2019).
- 8) Yang, F., Kou, J., Liu, Z., Li, W. & Du, W. MYC Enhances Cholesterol Biosynthesis and Supports Cell Proliferation Through SQLE. *Frontiers in Cell and Developmental Biology* **9**, (2021).
- 9) Colic, M. *et al.* Identifying chemogenetic interactions from CRISPR screens with drugZ. *Genome Med* **11**, 52, (2019).
- 10) Love, M. I., Huber, W. & Anders, S. Moderated estimation of fold change and dispersion for RNA-seq data with DESeq2. *Genome Biol* **15**, 550, (2014).
- 11) Mellman, I., Chen, D. S., Powles, T. & Turley, S. J. The cancer-immunity cycle: Indication, genotype, and immunotype. *Immunity* **56**, 2188-2205, (2023).
- 12) Amodio, V. *et al.* Genetic and pharmacological modulation of DNA mismatch repair heterogeneous tumors promotes immune surveillance. *Cancer Cell* **41**, 196-209.e195, (2023).

- 13) Sun, X. *et al.* Lanosterol synthase loss of function decreases the malignant phenotypes of HepG2 cells by deactivating the Src/MAPK signaling pathway. *Oncol Lett* **26**, 295, (2023).
- 14) Nguyen, T. P. *et al.* Selective and brain-penetrant lanosterol synthase inhibitors target glioma stem-like cells by inducing 24(S),25-epoxycholesterol production. *Cell Chem Biol* **30**, 214-229 e218, (2023).
- 15) Phillips, R. E. *et al.* Target identification reveals lanosterol synthase as a vulnerability in glioma. *Proc Natl Acad Sci U S A* **116**, 7957-7962, (2019).
- 16) Maione, F. *et al.* The cholesterol biosynthesis enzyme oxidosqualene cyclase is a new target to impair tumour angiogenesis and metastasis dissemination. *Sci Rep* **5**, 9054, (2015).
- 17) Rao, C. V., Newmark, H. L. & Reddy, B. S. Chemopreventive effect of farnesol and lanosterol on colon carcinogenesis. *Cancer Detect Prev* **26**, 419-425, (2002).
- 18) Mafuvadze, B., Liang, Y. & Hyder, S. M. Cholesterol synthesis inhibitor RO 48-8071 suppresses transcriptional activity of human estrogen and androgen receptor. *Oncol Rep* **32**, 1727-1733, (2014).
- 19) Liang, Y. *et al.* Cholesterol biosynthesis inhibitors as potent novel anti-cancer agents: suppression of hormone-dependent breast cancer by the oxidosqualene cyclase inhibitor RO 48-8071. *Breast Cancer Res Treat* **146**, 51-62, (2014).

REVIEWERS' COMMENTS

Reviewer #1 (Remarks to the Author):

The authors have meticulously revised the manuscript, and its quality and novelty are at a satisfactory level for publication.

Reviewer #2 (Remarks to the Author):

The authors have addressed my concerns sufficiently.

Reviewer #3 (Remarks to the Author):

The authors address my concerns.

Reviewer #4 (Remarks to the Author):

In the revised manuscript, the authors have addressed most of my comments/concerns. However, I still have the following concerns:

1. In Supplementary Fig. 1g, the quality of the western, especially the DUSP18 western in B16-OVA cells, was too poor to evaluate the changes of the protein levels in control and shDUSP18 cells.
2. In Supplementary Fig. 1n, the percentage of CD4⁺ and CD8⁺ cells in response to anti-IgG and anti-CD8 α was shown. In the sample with anti-IgG treatment, 34.7% is CD4⁺ cells and 21% is CD8⁺ cells. In the sample treated with anti-CD8 α , 41.1% is CD4⁺ cells and 1.2% is CD8⁺ cells. How these percentages were calculated? Is the rest of the ~50% cells the double negative cells? If so, this population does not look like have ~50%?
3. In Supplementary Fig. 2g, how the IFN γ ⁺ and GzmB⁺ cells were calculated need to be stated in the figure legend?
4. In Supplementary Fig. 2K, it looks like that the cyclin D1 was increased in CKO. Could the authors provide a quantification of the western results?

Point-by-point response to reviewers' comments (NCOMMS-24-07775A)

First of all, we sincerely appreciate the four reviewers spending time on handling and reviewing our revised manuscript.

Reviewer #1 (Remarks to the Author)

The authors have meticulously revised the manuscript, and its quality and novelty are at a satisfactory level for publication.

Response: We thank the reviewer for the positive and encouraging comments.

Reviewer #2 (Remarks to the Author)

The authors have addressed my concerns sufficiently.

Response: We thank the reviewer for supportive comments.

Reviewer #3 (Remarks to the Author)

The authors address my concerns.

Response: We are grateful for the positive comments provided by our reviewer.

Reviewer #4 (Remarks to the Author)

In the revised manuscript, the authors have addressed most of my comments/concerns. However, I still have the following concerns:

Response: We sincerely thank the reviewer for the excellent suggestions to guide us to further improve our manuscript and agree that we have addressed most of the reviewer's comments.

1. Comment: In Supplementary Fig. 1g, the quality of the western, especially the DUSP18 western in B16-OVA cells, was too poor to evaluate the changes of the protein levels in control and shDUSP18 cells.

Response: As suggested, the high quality of the western blots was provided in **Supplementary Fig. 1g** of the revised manuscript.

2. Comment: In Supplementary Fig. 1n, the percentage of CD4⁺ and CD8⁺ cells in response to anti-IgG and anti-CD8 α was shown. In the sample with anti-IgG treatment, 34.7% is CD4⁺ cells and 21% is CD8⁺ cells. In the sample treated with anti-CD8 α , 41.1% is CD4⁺ cells and 1.2% is CD8⁺ cells. How these percentages were calculated? Is the rest of the ~50% cells the double negative cells? If so, this population does not look like have ~50%?

Response: We thank the reviewer for pointing out this critical concern. This misinterpretation is due to the fact that our graphical analysis resulted in many cells being on the chart edges of the picture. To address this concern, we have re-analyzed and presented this data in **Supplementary Fig. 1n** of the revised manuscript. The percentage of CD8⁺ and CD4⁺ T cells is calculated by calculating the proportion of these cells among the T cells in the spleen.

3. Comment: In Supplementary Fig. 2g, how the IFN γ ⁺ and GzmB⁺ cells were calculated need to be stated in the figure legend?

Response: Following your suggestion, the corresponding calculation has been stated in the figure legend in our revised manuscript.

4. Comment: In Supplementary Fig. 2K, it looks like that the cyclin D1 was increased in CKO. Could the authors provide a quantification of the western results?

Response: As suggested, we have provided the quantification of the western results in **Supplementary Fig. 2k** of the revised manuscript.